# Recent trends of groundwater temperatures in Austria

Susanne A. Benz[1], Peter Bayer[2], Gerfried Winkler[3], Philipp Blum[1]

[1] Institute of Applied Geosciences (AGW), Karlsruhe Institute of Technology (KIT), Karlsruhe, 76131, Germany
[2] Institute of new Energy Systems (InES), Ingolstadt University of Applied Sciences, Ingolstadt, 85019, Germany
[3] Institute of Earth Sciences (IEW), NAWI Graz Geocenter, University of Graz, Graz, 8010, Austria

*Correspondence to*: Susanne Benz (susanne.benz@kit.edu)

## Abstract

Climate change is one if not the most pressing challenge modern society faces. Increasing temperatures are observed all over the planet and the impact of climate change on the hydrogeological cycle has long been shown. However, so far we have insufficient knowledge on the influence of atmospheric warming on shallow groundwater temperatures. While some studies analyse the implication climate change has on selected wells, large scale studies are so far lacking. Here we focus on the combined impact of climate change in the atmosphere and local hydrogeological conditions on groundwater temperatures in 227 wells in Austria, which have in part been observed since 1964. A linear analysis finds a temperature change of + 0.7 ± 0.8 K in the years from 1994 to 2013. In the same timeframe surface air temperatures in Austria increased by 0.5 ± 0.3 K displaying a much smaller variety. However, most of the extreme changes in groundwater temperatures can be linked to local hydrogeological conditions. Correlation between groundwater temperatures and nearby surface air temperatures was additionally analysed. They vary greatly with correlation coefficients of -0.3 in central Linz to 0.8 outside of Graz. In contrast, the correlation of nationwide groundwater temperatures and surface air temperatures is high with a correlation coefficient of 0.83. All of these findings indicate that while atmospheric climate change can be observed in nationwide groundwater temperatures, individual wells are often primarily dominated by local hydrogeological conditions. In addition to the linear temperature trend, a step-wise model was also applied that identifies climate regime shifts, which were observed globally in the late 70s, 80s, and 90s. Hinting again at the influence of local conditions, at most 22 % of all wells show these climate regime shifts. However, we were able to identify an additional shift in 2007, which was observed by 37 % of all wells. Overall, the step-wise representation provides a slightly more accurate picture of observed temperatures than the linear trend.

## 1 Introduction

The thermal regime in the ground is coupled with the conditions in the atmosphere, and air temperature variations leave their traces in the ground. While, already at depth of a few meters, the amplitudes of periodic diurnal and seasonal temperature trends are strongly attenuated (Taylor and Stefan, 2009), long term non-periodic changes of air temperature permanently influence the subsurface down to greater depths of several tens to hundreds of meters (Beltrami et al., 2005). Worldwide, borehole temperature profiles therefore witness the increase of surface air temperature (SAT) due to recent climate (Huang et al., 2000; Harris and Chapman, 1997). In borehole climatology,

focus is set on "dry" boreholes in undisturbed natural areas, that is, boreholes with negligible influence of
groundwater flow and no direct human impacts. Borehole temperatures logged in such boreholes can be used to
invert vertical conductive heat transport models for deriving the corresponding trend of ground surface temperature
(GST). By assuming that GST and SAT are directly coupled or similar, past climate can be reconstructed. Many
boreholes, however, are located in urbanized areas and regions with past changes of land cover, where often
accelerated ground heat flux and higher GST are observed (Bense and Beltrami, 2007; Menberg et al., 2013; Bayer et
al., 2016; Cermak et al., 2017). Moreover, in humid climate regions boreholes are mostly not dry, but drilled for
groundwater use or monitoring. When dynamic groundwater flow conditions exist, then advective heat transport can
substantially affect the thermal regime in the subsurface (Ferguson et al., 2006; Kollet et al., 2009; Taylor and
Stefan, 2009; Stauffer et al., 2017; Westaway and Younger, 2016; Uchida et al., 2003). Additionally, recharge
processes, including snowmelt and rain-derived recharge, might impact the thermal regime of the shallow
subsurface. Previous studies, however, indicate that in many cases their influence can be neglected. Ferguson and
Woodbury (2005) and Bense and Kurylyk (2017) demonstrated that it is possible to estimate groundwater recharge
by using temperature-depth profiles based on the common assumption that the mean annual groundwater recharge
temperature is equal to the mean annual surface air temperature. Menberg et al. (2014) showed in their study that the
contribution of snowmelt-induced recharge with low temperature is minor in comparison to the overall recharge.
Finally, Molina-Giraldo et al. (2011) investigated the impact of seasonal temperature signals into an aquifer upon
bank infiltration including also varying groundwater recharge temperatures. They showed that the convective heat
transfer by groundwater recharge compared to conduction through the unsaturated zone and convection within the
aquifer is of minor impact. Still, the interplay of long-term climate variations, land use change and groundwater
produces a complex transient system, which is difficult if not impossible to accurately understand based on a few
borehole measurements (Irvine et al., 2016; Kupfersberger et al., 2017; Kurylyk et al., 2017; Kurylyk et al., 2014;
Kurylyk et al., 2013; Taniguchi and Uemura, 2005; Taniguchi et al., 1999; Zhu et al., 2015).
The consequence of climate change on aquifers was illuminated with respect to groundwater recharge and
availability of freshwater resources (Moeck et al., 2016; Scibek and Allen, 2006; Holman, 2006; Gunawardhana and
Kazama, 2011; Loáiciga, 2003), groundwater quality impacts (Kolb et al., 2017) and effects on groundwater (-
dependent) ecosystems (Burns et al., 2017; Jyväsjärvi et al., 2015; Kløve et al., 2014; Andrushchyshyn et al., 2009;
Hunt et al., 2013). Taylor et al. (2012) summarized various connections and feedbacks between climate change and
groundwater. A key parameter is the temperature, which is expected to increase in shallow groundwater globally
following with some delay following roughly the trends in the atmosphere. However, long-term measurements of
temperature evolution in groundwater are rare (Watts et al., 2015; Figura et al., 2015). Instead often well
measurements taken at a few different time points are compared to indicate elevated temperatures, such as by
Gunawardhana and Kazama (2011) for the Sendai Plain in Japan, by Šafanda et al. (2007) for boreholes in the Czech
Republic, Slovenia and Portugal, and Yamano et al. (2009) and Menberg et al. (2013) for urban areas in Eastern Asia
and Central Europe. Others, such as Kupfersberger (2009) and Menberg et al. (2014) examine repeated temperature
records of single or a few selected wells. The work by Lee et al. (2014) is one of the very few studies on long term
groundwater temperature (GWT) time series recorded for a larger area. They applied linear regression to hourly
temperature data recorded from 2000 to 2010 at 78 South Korean national groundwater monitoring sites. They found
a mean increase of 0.1006 K/year and concluded that shallow ground and surface temperature show moderate
proportionality. Lee et al. (2014), however, reported that 12 wells revealed decreasing GWT trends without further
details on potential factors. Blaschke et al. (2011) applied trend analyses on long term data sets of mean annual GWT
of 112 and 255 wells for the time periods 1955-2006 and 1976-2006 respectively in Austria. They found increasing
trends of the GWT in shallow porous aquifers related to increasing air temperature. Similar insights from other
regions are lacking still, and the contribution of atmospheric warming to long-term GWT evolution is nearly
unexplored.
In the presented study, GWTs of 227 wells in Austria, measured in part since 1966, are analysed and regional
patterns and temperature anomalies are identified. In contrast to Blaschke et al. (2011) focus here is not only set on
linear trends, but also on detection of climate regime shifts in the measured GWT, following the suggestions by
Figura et al. (2011) and Menberg et al. (2014). As a relevant mode of global climate variability, long-lived decadal
patterns such as the Atlantic or Pacific decadal oscillation have been identified, e.g. Minobe (1997) and Rodionov
(2004). These control atmospheric temperatures as well and are often described as sudden, step-wise temperature
changes separating stable periods, called climate regimes. Even if these regime shifts arrive attenuated and delayed
in shallow groundwater, they can be detected and thus can offer another hint on the influence of climatic variations.
Aside from the statistical analysis of GWT time series, the influence of land cover as well as their correlation to
surface air temperature is investigated to scrutinize potential local influences on the measured data.

## 2 Material and Methods

### 2.1 Material

**Geology, Hydrogeology and Climate of Austria**

The Austrian Alps as the main part of the European Eastern Alps are characterized by a complex geology with
various lithologies and have been built up during multiple tectonic phases striking now in a West-East direction. The
complexity of the tectonic and geologic settings of the European Alps and in particular of the European Eastern Alps
is described and discussed by numerous authors (e.g. Schmid et al., 2004; Linzer et al., 2002). Active tectonic
evolution resulting in high topography and uplift rates coincide largely with high stream power (Robl et al., 2017;
Robl et al., 2008) and thus, have an impact on the drainage system of the Alps. During the Pleistocene the Alps were
affected by glaciations with a strong impact on the morphology in particular on the inner alpine valleys and the
foreland. Due to sedimentation during the Holocene these areas now contain quaternary porous aquifers. The herein
analysed wells are located in shallow aquifers representing these quaternary sediments in the inner alpine valleys and
foreland basin. Based on a compiled geology a hydrogeological overview as a hydrogeological map of Austria is
provided by Schubert et al. (2003).
Climate and climate trends during the last two century (1800-2000) of the Great Alpine Region (European Alps and
their surrounding foreland, GAR) was intensively investigated during last decades yielding in the HISTALP data set
(Auer et al., 2007). This data set left its mark on the regional classification of climate zones by Köppen-Geiger where
Austria is mainly divided into three climate zones, warm temperate, boral, and alpine.

## Groundwater Temperatures

In Austria, GWTs up to Dec 2013 are provided by the Austrian Federal Ministry of Sustainability and Tourism Directorate-General IV. - Water Management (BMNT, former Federal Ministry of Agriculture, Forestry, Environment and Water Management (BMLFUW) in 1138 wells. Here, we focus on all wells with a measurement depth of less than 30 m, a record of at least 20 years and no major breaks (> 3 month) in the last 20 years of the time series. Hence, all studied wells are monitored at least since Jan 1994, and some already since 1966 (see Fig. S1a for more information). Additionally wells impacted by geothermal hot springs were excluded. Overall, in this study annual mean data of 227 individual wells from all over the country (Fig. 1a) are analysed. Years with less than 9 months of data are excluded. For the timeframe 1994 and 2013, this amounts to 74 excluded data points in 60 wells. Additionally, only 9-11 months of data were available for 260 data points in 122 wells. To minimize the associated bias, these small gaps in the time series were filled using a linear fit. Hence small errors for years without a full set of monthly mean data have to be expected.

The average measurement depth in the wells is $7 \pm 4$ m below ground surface (Fig. S1b). All wells are located in the Cfb climate zone of the Köppen-Geiger classification, warm temperate climate with warm summers and no dry seasons (Rubel et al., 2017). The spatial median GWTs and inner 90 % percentiles for all wells are displayed in Fig. 1b. The obtained temperature in Fig. 1b increases from around 9.8 °C in 1966 to 11.4 °C in 2013.

Following the CORINE Land Cover (CLC) data from 2012 (Fig. S2a), 45 % of all wells are under artificial surfaces, 46 % under agricultural areas, and 9 % under forest following the 100 m × 100 m classification. In addition, CLC from 1990 was consulted, however, no land cover changes near any of the analysed wells are observed. Overall, for the time period 1994 – 2013, absolute GWTs of the monitored wells under artificial surfaces are on average $1.5 \pm 0.3$ K warmer than GWTs under forest; GWTs under agricultural areas are on average $0.6 \pm 0.2$ K warmer than GWTs under forest (Fig. S2b). This validates previous findings by Benz et al. (2017b) for GWTs in Germany, who identified even larger differences of up to 3 K between the individual land cover classes.

## Surface Air Temperatures

Surface air temperatures (SATs) within Austria are monitored by the Central Institution for Meteorology and Geodynamics (ZAMG), Austria. In this study data from 12 individual weather stations are being analysed, each one is located within 5 km of at least one analysed well and in the same climate zone (Cfb). Their location is displayed in Fig. 1a. Again annual mean data was available for a time period of 1966 to 2013 (Fig. 1b). As expected and as previously shown in Benz et al. (2017b) for SAT and Benz et al. (2017a) for land surface temperatures, above ground temperatures are generally lower than GWTs. All 12 analysed weather stations are located in areas classified as artificial surface and experienced no land cover changes.

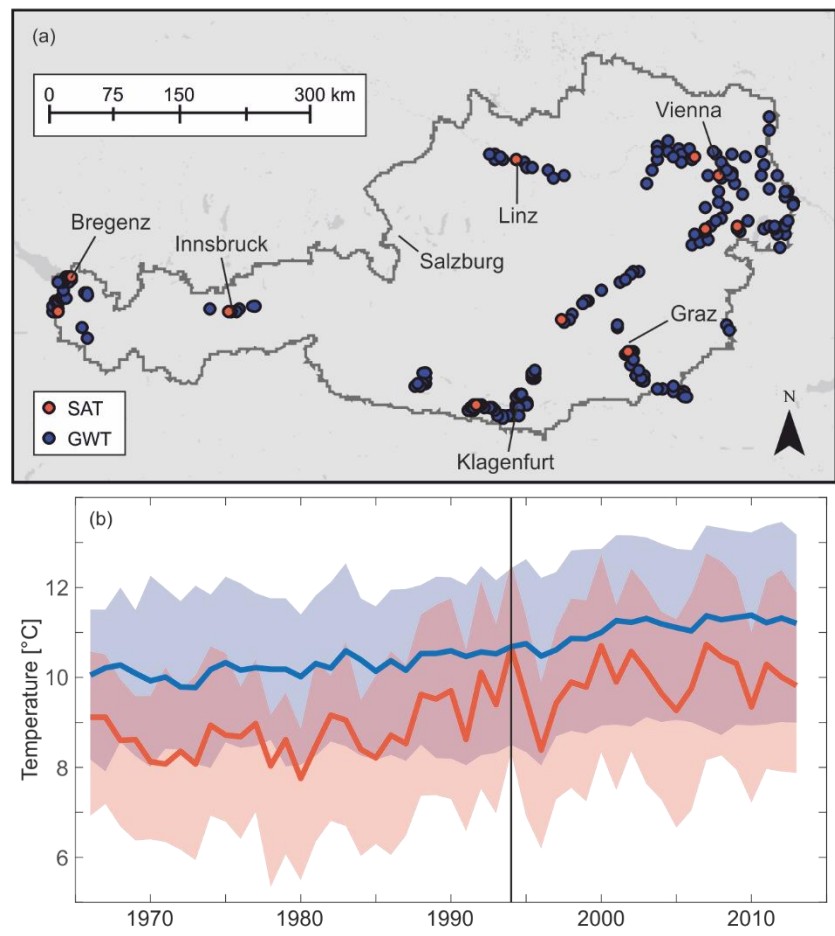

139

**Figure 1. (a) Location of all analysed groundwater temperature (GWT - 227 wells) and surface air temperature (SAT - 12 weather stations) measurement points; (b) temporal evolution of the spatial median, annual mean temperatures for groundwater (blue) and air (red). The inner 90 percentiles are marked in lighter colours. All time series were monitored since at least 1994.2.2 Method**

**Correlations**

Within this study, the Spearman correlation coefficient was determined, as it is especially robust to outliers caused for example by heat waves, which impact air temperatures but have only minor effect on groundwater temperatures. When determining the correlation between two time series, missing years were ignored. Next to the correlation between GWT and SAT, correlation between all individual wells and weather stations were determined in order to create a plot similar to a (semi)variogram that shows the correlation between two measurement stations depending on their distance to each other.

**Linear analysis**

Equivalent to the work by Lee et al. (2014), a linear temperature change was determined for all 227 wells. For this, a linear regression model of the annual mean temperature data was determined in Matlab 2016b. Because all wells in our dataset were continuously monitored between 1994 and 2013, only this timeframe was analysed.

**Climate regime shifts**

Climate data is often thought to not change linearly, but in form of a step function, dividing a time series into individual climate regimes of a constant mean (Andrushchyshyn et al., 2009; Minobe, 1997). These regimes are changed when so-called climate regime shifts (CRS) occur and long-term mean values change. While several methods to model these shifts have been in use (Easterling and Peterson, 1995), in recent years the method by Rodionov (2004) became standard. It identifies the significance of each possible shift by calculating the so-called Regime Shift Index (RSI): the cumulative sum of the normalized differences between the observed values and the long term mean of the assumed regime. Only shifts with a positive RSI are considered significant, and a higher value of RSI denotes a more pronounced CRS. The entire algorithm is described in detail by Rodionov (2004). This sequential analysis is data driven and requires no prior knowledge of the timing of possible shifts. It was updated to further include prewithening in order to reduce background noise (Rodionov, 2006) and is available online as a Microsoft Excel add-in (NOAA). In this study we applied the method to the complete time series of all 227 wells and 12 weather stations. Because the algorithm cannot handle gaps within the analysed series, gaps in our data were filled using a linear fit. All parameters were set to the same values as in the work by Menberg et al. (2014), who applied the method to four GWT time series in Germany. A target significance level of 0.15 was used by Menberg et al. and in our analysis, the cut off length was set to 10 years and the Huber weight parameter was set to 1.

**3 Results and Discussion**

**3.1 Correlations**

Figure 2a displays the correlation between different wells or rather different weather stations in relation to their distance to each other. Shown is the distance between two wells/weather stations on the x-axis and the corresponding spearman correlation coefficient between them. For the weather station, each individual pair is shown by a red point, for GWTs, as there are many possible pairs of wells, the line gives the moving median (± 25 km) correlation of all pairs at the corresponding distances. The inner 90 percentiles are shown in grey, and correlation coefficients close to or below zero are determined for several pairs of wells. However, here p-values are generally also close to one and GWTs do not correlate. This is most likely due to local heat sources impacting at least on well in these pairs.

As expected the moving average correlation decreases with distance. This decrease is more extreme in GWTs than in SATs and GWTs correlate less than SATs overall. This agrees with the observations in Benz et al. (2017b), who showed that annual mean GWTs show greater variations than SAT over the same distances.

Additionally, the correlation between two wells seems to be anisotropic: correlation coefficients between two wells decrease faster with north-south distance than with west-east distance (Fig. 2b), which can be explained by the dominant striking direction of the geology and the resulting topography in Austria, where valleys generally run from west to east. Hence, larger rivers typically follow this direction and wells at the same latitude experience similar temperature signals.

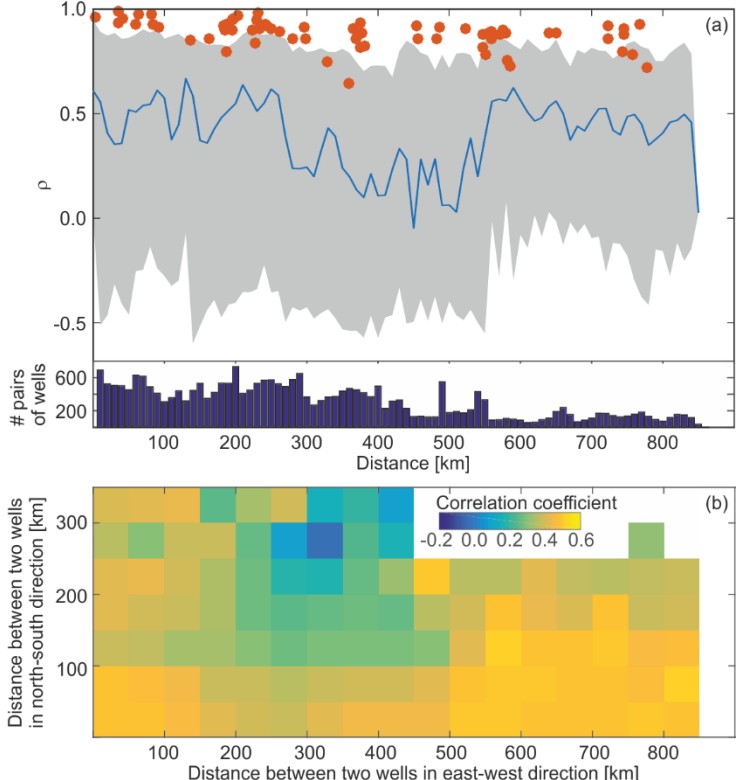

188

**Figure 2. Influence of distance on the correlation between the annual means of two measurement points. a) Correlation between SAT time series is given in red, median correlation between GWT time series is given in blue. The inner 90 percentile are coloured in grey, the number of pairs of wells per distance is shown in dark blue below. b) The colour gives the median correlation between GWTs of two wells in relation to their absolute distance to each other in east-west direction (x-axis) and in north-south direction (y-axis).**

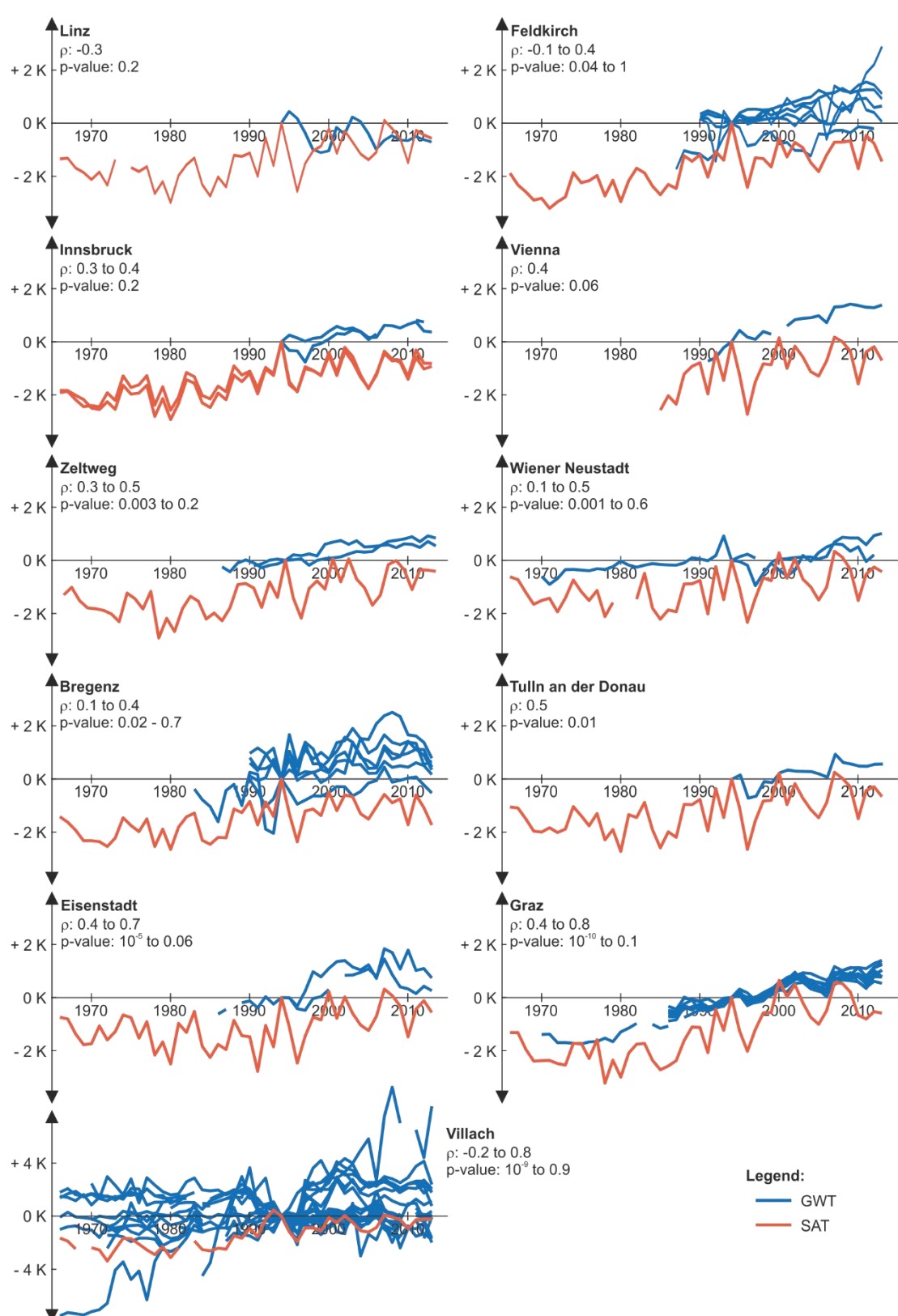


**Figure 3. Change from 1994 in surface air temperature (SAT) and groundwater temperatures (GWTs) of all wells within 5 km of the analysed weather station. See Fig. S3 for an overview of the locations. Minimum and maximum correlations and p-values between individual wells and weather stations are given.**

In a next step, correlations between GWT and SAT are determined. On a country-wide scale Spearman correlation
coefficient between spatial median GWT and SAT (Fig. 1b) is 0.83. In comparison correlation between individual
weather stations and wells are shown in Fig. 3, locations are displayed in detail in Fig. S3. Here correlations vary
greatly and Spearman correlation coefficients are < 0.5 for about half of all wells within 5 km of a weather station.
This indicates that GWTs are often influenced by local causes and not necessarily solely by local SATs. The lowest
correlation is determined in Linz where the groundwater is intensively used for cooling and heating (Krakow and
Fuchs-Hanusch, 2016). The studied well is located within the city centre next to train tracks and office buildings.
Hence, it is very likely that the thermal properties of the groundwater are dominated by anthropogenic influences
from heated buildings and underground structures as often the case in subsurface urban heat islands (Menberg et al.,
2013, 2013; Benz et al., 2015; Benz et al., 2016; Attard et al., 2016). This would also explain the high GWTs that are
on average 3.3 K warmer than the local annual mean SAT. Like the well, the weather station is also located within
the city centre.
The best correlations between individual pairs of a well and a weather station can be observed in the southern part of
the city of Graz, where all wells and the weather station are located close to or within the Graz airport, respectively.
The well with the highest correlation of 0.80 to SAT is located less than 1 km from the weather station close to the
airport parking lot next to suburban housing. It is continuously monitored since 1970 and the longest time series in
the area. The well with the lowest correlation (0.45) to the weather station here is located slightly to the east near a
dog-park and suburban housing. Here observations started in 1994, it is the shortest time series in this area. At all
other wells, measurements began in 1986 and show correlations between 0.6 and 0.7 to SAT indicating that the
duration of the measurements play a significant role for local comparisons. In contrast, duration of the time series
appears to be of minor importance on a countrywide scale. For example, the long time series in Wiener Neustadt
(Fig. 3), which started measurements in 1970 and is located near a mineral extraction site, has a correlation of 0.48
and is therefore comparable to the short time series in Graz, starting in 1994 located in a suburban area.
Additionally, measurement depths of GWT can have an impact on the correlation between SAT and GWT. While it
is generally assumed that a measurement depth closer to the surface results in a better correlation with SAT as there
is less of a shift between both datasets, this is only the case for some of the here analysed locations such as Villach
(Figure S4a). In contrast, correlation increases with GWT measurement depth for other locations such as the one in
Graz. This might be related to local underground heat sources such as sewage systems impacting GWT near the
surface more than temperatures at greater depth. However, as the depth of the wells analysed here varies only
slightly, no definite conclusions can be drawn without further inspection of specific cases.

**Table 1. Correlation coefficient and corresponding p-value between spatial median SAT and spatial median GWT for all analysed SAT locations, and additional information.**

| Location | Number of wells | Measurement depth GWT [m below surface] | Number of weather stations | Spearman correlation | p-value | Population[1] |
|---|---|---|---|---|---|---|
| Linz | 1 | 10 | 1 | -0.31 | $10^{-1}$ | 192,000 |
| Feldkirch | 6 | 4 to 17 | 1 | 0.19 | $10^{-1}$ | 31,000 |
| Innsbruck | 2 | 10 | 2 | 0.37 | $10^{-1}$ | 123,000 |
| Vienna | 1 | 12 | 1 | 0.41 | $10^{-2}$ | 1,740,000 |
| Zeltweg | 2 | 6 to 7 | 1 | 0.48 | $10^{-3}$ | 7,000 |
| Wiener Neustadt | 2 | 9 to 20 | 1 | 0.51 | $10^{-4}$ | 42,000 |
| Bregenz | 6 | 4 to 10 | 1 | 0.52 | $10^{-3}$ | 28,000 |
| Tulln an der Donau | 1 | 7 | 1 | 0.54 | $10^{-2}$ | 15,000 |
| Eisenstadt | 2 | 4 to 5 | 1 | 0.67 | $10^{-4}$ | 13,000 |
| Graz | 9 | 4 to 12 | 1 | 0.73 | $10^{-8}$ | 266,000 |
| Villach | 17 | 3 to 11 | 1 | 0.80 | $10^{-11}$ | 60,000 |

[1] Register-based Labour Market Statistics 2014, municipality level (Statistik Austria).

Table 1 displays the correlations between spatial median GWT and spatial median SAT for each of the SAT locations in Figs. 3 and S3. For all locations with at least two wells besides Zeltweg and Graz correlation does improve when spatial median GWT is analysed instead of the individual locations. In all likelihood the spatial median GWT provides a more general temperature trend that is not influenced by local influences on temperatures such as construction work, plant development and shading, and is therefore more closely related to surface air temperatures.

In addition, the data indicates that city size or rather population of the city does not necessarily influence the correlation between GWT and SAT (Table 1). For example, both locations Graz (population of more than 250,000) and Eisenstadt (population of 13,000) have similar correlation coefficients despite their different population. Meanwhile, Bregenz and Feldkirch have a similar population (~30,000) and number of wells (six), but different correlation coefficients (0.52 and 0.19). However, it is also important to note that not all wells analysed here are located in the city centre, still all of them are within close proximity (< 250 m) of anthropogenically used areas (Fig. S3).

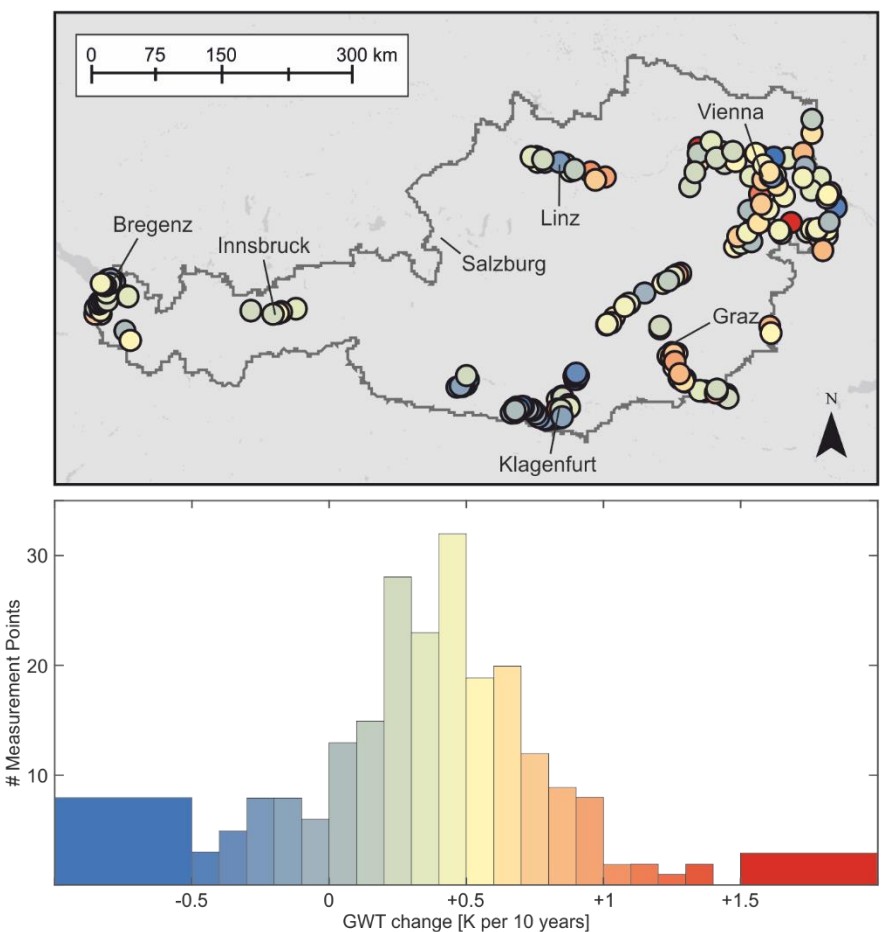

244

**Figure 4. Increase in temperature for all individual measurement points for the 20-year timeframe 01/1994 to 12/2013. The mean change in groundwater temperature is +0.4 ± 0.5 K per 10 years.**

**3.2 Linear temperature change**

During the time between 1994 and 2013, GWTs have changed on average by +0.36 ± 0.44 K per 10 years and SAT on average by +0.24 ± 0.13 K per 10 years. The lower changes in SAT are most likely due to the chosen timeframe: A heat wave in summer 1994 led to extraordinary high annual mean SAT in this year (Figure 1b) and thus impacts the determined linear temperature change. The increase of GWT is in good agreement with results of a former study considering data sets of Austria from 1976-2006 (Blaschke et al., 2011). However, it is more than double the global air temperature increase determined by Jones et al. (1999) for the timeframe 1978 to 1997 with +0.32 K in 20 years and less than the numbers given in the work by Ji et al. (2014). In their global study they give an air temperature increase of more than 0.4 K for the timeframe 2000 to 2009 for the northern mid-latitudes including Austria. Fig. 4 displays a map and a histogram of all determined GWT changes. There appears to be no significant influence of land cover on the observed temperature change (Fig. S2c). Median temperature change is approximately 0.4 ± 0.4 K per 10 years for groundwater under artificial surfaces and forest areas, and 0.3 ± 0.5 K per 10 years under cultivated areas. However, temperature change decreases slightly with GWT measurement depth by approximately 0.015 K per 10 years per meter (Fig. S4b). This relationship can be related to deeper temperatures corresponding to earlier temperatures, when temperature increase was less severe. However, because the vast majority of temperatures are

monitored at a depth of less than 15 m and show a high variability in linear temperature change, this number must be
taken with caution. $R^2$ of the fit is only 0.02 and RMSE is 0.4 K.
To evaluate the goodness of this linear approach when representing climate change, RMSE of the fit was determined
for each well for 1994 to 2013. We found an average RMSE of $0.4 \pm 0.2$ °C.
When looking at the individual wells, no obvious spatial pattern for temperature changes is visible (Fig. 4). However,
most wells with temperature changes lower than the $5^{th}$ percentile are located close to the river Drava in Ferlach,
Villach, and Kleblach-Lind in the very South of Austria (Fig. 5 and Fig. S5). Although, they are up to 80 km away
from each other, all of these wells show a sudden drop in temperatures in the year 2007 (wells Ia, Ib, IIa, IIb, Va, and
Vb marked blue in Fig. 5). This temperature reduction can be seen in most of the 27 wells that are less than 1 km
from the Drava away (Fig. S6), for 24 of these wells, temperatures in 2006 are more than 0.6 K warmer than
temperatures in 2008. However, temperatures (as well as additional parameters such as water level) within the river
do not indicate any connection between this sudden temperature reduction and the Drava river (Fig. S6). Either way,
further research is necessary to identify the cause of this temperature anomaly.
Additionally, three other wells in the lowest 5 percent of temperature change are all located less than 10 km from
each other near the village Kappel am Krappfeld (wells IVa, b and c marked orange in Fig. 5). They and also
additional surrounding wells show a steep decline in temperatures in 2006 before temperatures start to increase
steadily again. These wells seem to be affected by the new drinking water supply (four wells with a total pumping
rate of about 100 l/s) located about 1 km in the south. This demonstrates the importance of including groundwater
flow when trying to interpret groundwater temperature. In general, most of the extreme changes in temperature
appear to be linked to local causes and do not happen gradually, but rather rapidly over the short time span of one or
two years. Another example of this can be seen in wells with temperature changes higher than the $95^{th}$ percentile
(Fig. 5 and Fig. S7). While these highest five percentiles of all wells do not show local clusters to the same extent as
the lowest 5 percentiles and can be observed all over the country, three wells (1a, 1b and 1c, marked dark blue in Fig.
5) are located in the industrial area of Villach in the South of Austria. Here some construction work during 1997 is
likely the cause of the sudden temperature increase but concrete evidence could not be identified.

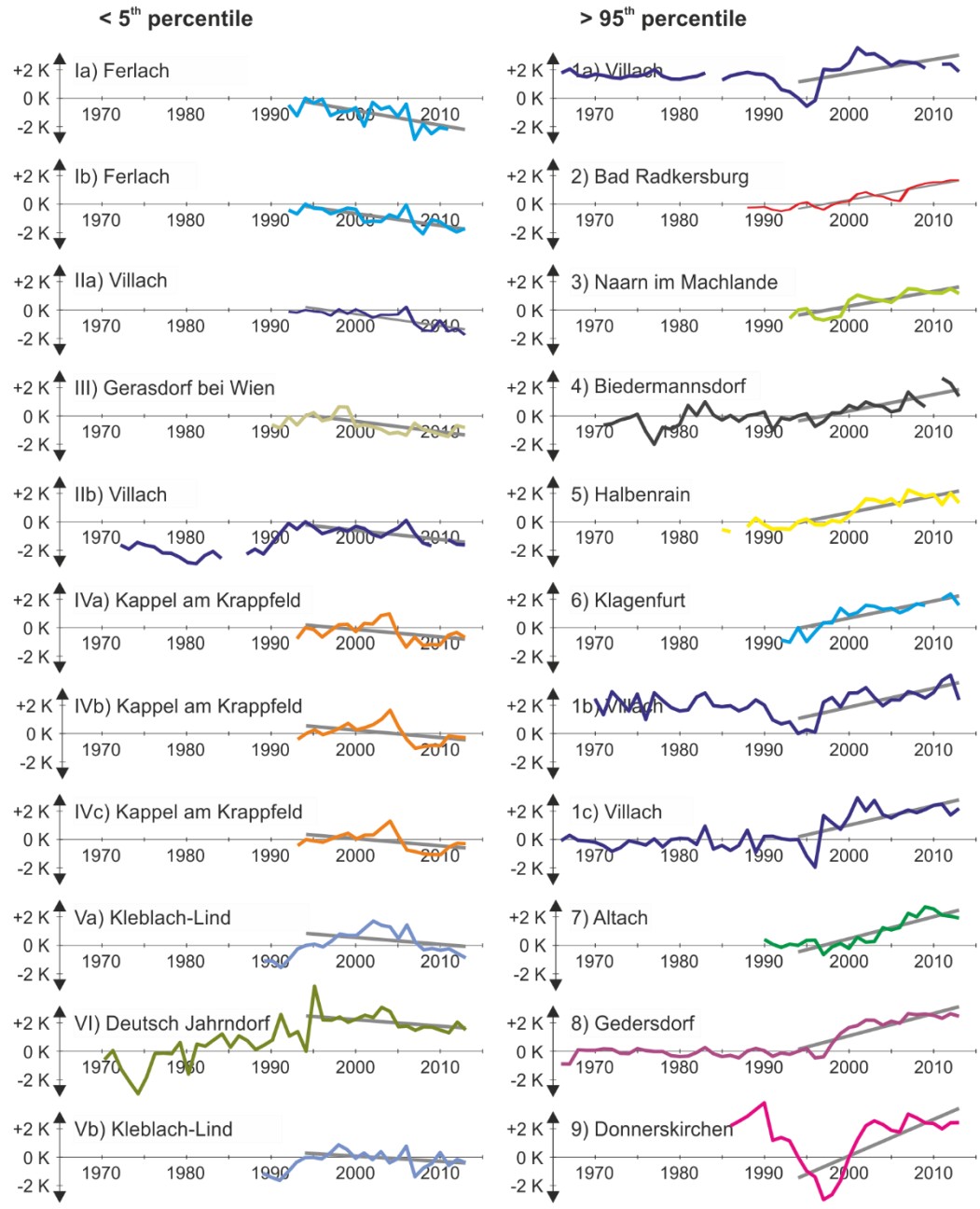


**Figure 5. Annual mean time series and linear fit (in grey) of the wells with the lowest (left side, numbered with roman numbers) and highest (right side, numbered with arabic numbers) temperature changes in the time frame 1994 and 2013. See Fig. S5 and S7 for an overview of the locations. They are placed in ascending orders with the highest temperature change at the bottom.**

### 3.3 Climate regime shifts

All detected climate regime shifts (CRS) of the spatial median temperatures time series are shown in Fig. 6a. Overall GWTs increase by 1.2 K between the first and last CRS and SAT increased by 1.5 K.

Global climate regime shifts (CRS) in air and also groundwater were detected for the late 70s, the late 80s and the late 90s by Menberg et al. (2014). Using the same algorithm spatial median annual mean GWT and SAT in Austria show shifts in the late 80s and 90s (Fig. 6a). GWTs show additional shifts in 1981 and 2007. While the shift in the

late 80s is observed during the same year (1988) in GWT and SAT, the shift in the late 90s appears earlier and is
more significant in GWTs. However, because SATs are the drivers of GWTs and not vice versa, the fact that the
GWT change precedes the SAT change suggests that this method does not have the necessary resolution to determine
short time lags between SATs and GWTs. Accordingly the detected time shifts in wells within 5 km of a weather
station do generally not indicate the same CRS as the weather station: Of 56 CRS observed in at least one well only
12 are also observed in a nearby weather station no more than one year before (Fig. S8). However, it is also
important to note that some of the analysed time series only span over a 20 year time period and are thus on the
shorter end for a statistically relevant analysis of climate regime shifts (Rodionov, 2006).

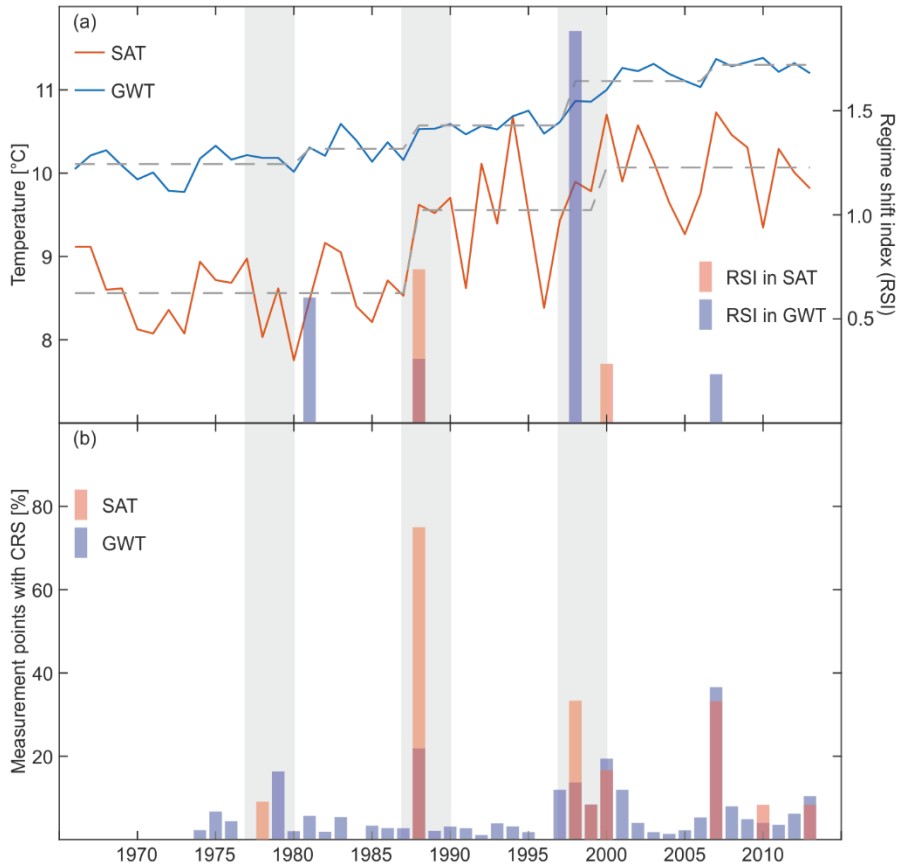


**Figure 6. (a) Median groundwater temperature (blue) and surface air temperature (red) of all wells or rather weather**
**stations as well as the corresponding climate regime shifts (CRS) in form of the regime shift index (RSI). (b) Percentage of**
**measurement points in GWT (blue) and SAT (red) that show a CRS in each year. The analysis of global temperatures data**
**indicates a regime shift at the end of the 70s, the 80s and the 90s which are shown here in as grey bars.**
Like with the linear approach, the goodness of the CRS and corresponding statistical step model was evaluated by
determining the RMSE for the time period 1994 to 2013. We determined a mean RMSE value of $0.3 \pm 0.1$ K, which
is slightly better than the RMSE for the linear fit as determined above ($0.4 \pm 0.2$ K). Only 20 of the 227 analysed
wells have a better RMSE with the linear approach than the statistical step model of the CRS approach. Hence, we
conclude that the CRS method is slightly more appropriate to simulate temperature changes in groundwater than a
linear approach even for time periods as short as 20 years. However, when the individual wells and weather stations
are analysed (Fig. 6b), globally observed CRS can be identified in at most 22 % (1988) of all wells. Results further
show that the shift in the 90s is temporally more spread out than the shifts in the 70s and 80s in both GWT and SAT.
This indicates that this shift is less well defined and temperatures of the globe became more variable in their
temporal evolution. In accordance to this interpretation there is a higher percentage wells with a CRS in all years
after 1996 than before. Furthermore, more than one third of all weather stations and wells detect a shift in 2007
indicating this year as the start of a new climate regime within Austria. While a CRS in 2007 was not observed by
Menberg et al. (2014) whom studied earlier time series than here, this year was also identified by Litzow and Mueter
(2014) as the start of a new regime for both climate and biological indicators within the North Pacific Ocean.
However the dimension of the shifts do not always agree for all wells. For example, wells experiencing a shift
observed in 2007 include all wells along the Drau observed in Fig. 5 and S5, which show a sudden drop in
temperature for this year. In contrast, the countrywide time series in Fig. 6a indicates a positive shift in temperatures.

## 4 Conclusions

Temperatures in 227 shallow wells and 12 weather stations in Austria, monitored in part since 1966, were analysed
in this study. Linear temperature change was determined and revealed a general increase in temperature between the
years 1994 and 2013 of approximately +0.36 ± 0.44 K per 10 years in the groundwater and on average +0.24 ± 0.13
K per 10 years in the air. Most extreme changes in groundwater temperatures, especially temperature decrease, could
be linked to local causes such as the installation of a new drinking water supply that influences nearby groundwater
wells. This reveals the extent in which groundwater temperatures are dominated by local events, groundwater flow,
and the thermal properties of the surrounding. When solving local problems we can therefore not recommend relying
on average relationships valid on a nation scale. Accordingly correlation between annual mean groundwater
temperatures and nearby (< 5 km) air temperatures varies greatly from -0.3 in Linz to 0.8 near Graz. However, if
spatial median groundwater temperatures and surface air temperatures of all of Austria are compared, we found a
significant correlation of 0.83 demonstrating once more that groundwater temperatures are closely linked to surface
temperatures and therefore experience climate change. However, globally observed climate regime shifts in the late
70s, 80s and 90s could only be identified in approximately 20 % of all wells. Nevertheless, we were able to observe
another shift in 2007 in 37% of all wells and 33 % of all weather stations indicating this year as the possible start of a
new climate regime within the alpine region. However, further research dedicated to other climate parameters such
as permafrost and snowfall is necessary to validate these findings. Additionally, our observations made in Austria
should be transferred to similar regions in the world testing the transferability of the presented results. Overall
climate regimes represent measured temperature slightly better (RMSE: 0.3 ± 0.1 K) than the linear fit (RMSE: 0.4 ±
0.2 K).
**Acknowledgements**
We would like to thank Erich Fischer (BMNT, former BMLFUW) for information and data regarding groundwater
temperatures and Alexander Orlik (ZAMG) for information and data regarding surface air temperatures of Austria.
Furthermore, we would like to acknowledge the financial support for the first author by the portfolio project
"Geoenergy" of the Helmholtz Association of German Research Centres (HGF) and the support by Deutsche
Forschungsgemeinschaft and Open Access Publishing Fund of Karlsruhe Institute of Technology. A big thank you
also to one anonymous reviewer and to R. Hunt for their helpful and very constructive comments.

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
