# Peer review of "Recent trends of groundwater temperatures in Austria"

_Hydrology and Earth System Sciences, 2017_

## Referee Comment (RC1) · Anonymous Referee #1 · 12 Jan 2018

This paper addresses an important and interesting topic regarding the influence of atmospheric warming on groundwater temperature (GWT) in shallow systems. The authors used temperature records from 229 wells located in Austria and climatic data from weather stations installed nearby the wells. The positioning of the paper within the framework of studies devoted to the impact of climate change on hydrological system is well presented. The authors found that nationwide temperatures of groundwater increase and correlate statistically well with surface air temperature (SAT). Additionally, authors have used linear and step-wise models to describe the evolution of temperatures. Based on the step-wise approach (which seems to be more accurate than the linear model) the authors have identified that groundwater respond to climate regime shifts with sudden increase in temperature. This paper has been carefully prepared

and is well written. The conclusions will definitely trigger the attention of the scientific community and the readers of HESS. Nevertheless I believe that some points need to be clarified before publication.

General comments:

I. Some aspects of the methodology are not clear or absent. More details on how the 229 wells investigated in this study have been selected is required. More information regarding the type of sensors used to monitor GWT would be helpful to appreciate the quality of the data analyzed. More information regarding the regression approaches is also needed. How the shifts in regimes are determined in the step-wise model (mathematically speaking)? I also raise some additional points regarding the methodology in the specific comments.

II. I believe that there is a discrepancy between the original objective of the paper, which aims at highlighting impact of climate change at regional (country) scale (Line 12), and the description of potential local effects for (some) specific wells and locations. Indeed, the authors describe potential factors which could explain uncorrelated data locally. Local information that are made available to the reader are to my opinion not sufficiently detailed to support the arguments. The conclusions are consequently difficult to trust. I would recommend to separate the description of local factors from the result of the regional statistical analysis (which to my opinion constitutes the novelty of this study). The local impacts could be introduced in a separate discussion section. In this specific section, the authors could provide an exhaustive list of potential factors that could explain uncorrelated data along with some examples from specific sites to illustrate the hypothesis.

III. I believe that the conclusions of this paper could be strengthened by performing a more robust multivariate statistical analysis (Principal Component Analysis for example) considering more factors which might have an influence on GWT, integrating not only SAT but also geology, land cover evolution, water level variation, precipitation,

population dynamic, length of the temperature time series...

Specific comments:

Line 68: "...over decades". Please be more precise here.

Line 73: "... step-wise increases between the regimes". This is not clear to me. What regimes? Please clarify.

Figure 1 b. needs clarification. The presence of 3 curves is confusing. Could you, for example, make the inner percentile filled with transparent colors?

Line 98: How the wells have been selected? What proportion of wells has been excluded from the database? See general comment.

Line 128. Please clarify why you choose 1994 as initial time for fitting.

Line 129: Knowing which software you used is not informative here...

Line 132: Please justify the choice of using the Spearman correlation coefficient and provide references.

Line 133: Taking annual mean values calculated with 8 months of data only may introduce some bias... Considering only years with full year of data would be more robust to my opinion. Otherwise, please discuss the limitations in the text. It is also not clear why yearly averages are used in the correlation analysis while the linear regressions are performed on monthly mean temperature (Line 129).

Line 132-136: It would be interesting to perform complementary correlation analysis accounting for other parameters such as depth of the wells, depth to the water table, geology, vegetation and land use. This could be assess with multivariate methods such as PCA. This could add valuable picture of the factors influencing the results.

Line 145: "Breaks within the data were filled using linear fit". This is not clear... Please provide more information why you have to fill gaps for this analysis (and not for the

other analysis?).

Lines 160-164 and Figure 2a and b: This part require clarifications. As the authors stated, it seems that the shape of Austria (political boundary) might influence the results. Also the topography, with E-W strike orientation, might also have an influence. It is not so surprising that the correlation is better E-W that N-S (same latitude and orientation of topography). I am wondering if the figures are really informative. . .the decreasing correlation with distance in the figure a) is not obvious with the sharp increase at 550 km. . . Does this distance correspond to a decrease of the number of wells considered in the calculation?

Figure 3 is interesting but difficult to read. Would it be clearer if you display the relative change in temperature for all the wells? What are the p values here (not introduced in the text)?

Line 175 -176: To what coefficient are you referring to? The p values in the figure 3?

Lines 187-190: Here it seems that the length of the time series is critical in the interpretation of the correlation analysis. . . Please discuss this point.

Table 1: What does p-value mean here? Not introduced in the text or the caption. . ..

Lines 205-206: Reference to table is missing. It is actually not a big difference of correlation coefficient 0.36 vs 0.24. . . The comparison with population density is not obvious to me from these values. Please clarify. The influence of city center and development of urban area is actually critical. Could it be possible that the increases in temperatures are partly related to urban development? Identifying the correlation with such factors could be assessed with a multivariate correlation methods (PCA).

Line 214-215: This difference in average changes in temperature with higher values for GWT than SAT is surprising. . . Could it reflect the effect of urban development or other anthropogenic activities (pumping, injection, heating system. . .).

Line 226: Please provide a reference to the figure supporting the statement that spatial

pattern of temperature changes is visible. . .

Lines 226-235: Too few information are available on the effect of this flood event. What was the difference in temperature between the river and GW during the event? Did it cover the entire well area? Estimated volume? Please provide more information or I would recommend to remove this paragraph.

Line 236-249: It is somehow surprising and confusing how local effects are introduced again. . . I believe that it should be discussed in a dedicated section discussing potential hypothesis that may explain uncorrelated data with eventually some examples of local factors from specific sites as examples.

Line 247-249: Do you mean that the hot springs appeared suddenly?... I imagine that they were active before and constitute a constant temperature boundary. . .

Line 262: I do not understand what the authors mean by "spatial median annual mean". . . please clarify.

Lines 263 - 266: I am confused here. How do you explain that the shift in GWT occurs earlier than for the SAT? If the "CRS method (do you mean step-wise method) cannot be used to determine the precise timelag between GWT and SAT" why do you use it?

Technical corrections:

Line 29-31: Reference is missing.

Line 58: Reference style for Menberg et al. (2014).

Line 72: Check reference style.

Line 128: should be "Equivalent to the work by Lee et al. (2014)".

Labels of figure 2b could be changed by Northing and Easting.

Figure 6. Please add legends to your figures.

[Figure]

---

## Referee Comment (RC2) · R. Hunt (Referee) · 29 Jan 2018

Thank you for the opportunity to review "Recent trends of groundwater temperatures in Austria" by Benz, Bayer, Winkler, and Blum. I enjoyed reading the manuscript and appreciate the work it represents. I have outlined my specific primary suggestions for improvement below. I've also included minor comments, along with typographical suggestions as requested by the Journal. Only the primary comments rise to the level of serious consideration and response. The authors should feel free to contact me if anything is unclear at rjhunt@usgs.gov.

Specific/Primary Comments:

1) Overall manuscript: It strikes me that a focus on annual air temperature misses a

fundamental process important for this discussion. The temperature of the groundwater system reflects the temperature of groundwater recharge. Groundwater recharge, however, is variable over time, thus annual temperature changes are likely too coarse to capture the temperature effects of inter-annual recharge process. That is: snowmelt recharge will be near 0 degrees C; rain-derived recharge will be warmer. Perhaps there is a shift in recharge from less snowmelt to warmer rain sources that is driven by air temperatures. A groundwater recharge approach means that the simple relation of air temperature to groundwater temperature is more indirect, and this additional "noise" to the signal is perhaps why the correlations are not higher.

2) Section Groundwater Temperature/Figure S1: Similar to comment #1, groundwater basins have a residence time, with multiple ages and potential lags. There is an assumption that all groundwater reflects current air temperatures (e.g., line 269) but this may not be the case. Given the importance of other factors such as residence time, and the unsaturated zone buffering that dampens the climatic drivers, it seems worthwhile to include well statistics relating to:

- Depth to water table

- Well open interval

- Distance the well's open interval is below land surface

- Distance the well's open interval is below the water table

- Estimated position in the groundwater flow system (e.g., uppermost, middle, discharge; near groundwater divide versus near flow system end; urban versus rural agriculture versus forest; high elevation versus low elevation)

3) Lines 104-106: It seems that only focusing on annual averages may limit the applicability of the insights. For example, for cold water fisheries it is usually the temperatures in the late summer – late fall that are important.

4) Figure 2: The shaded area and short-duration blue line dipping below y=0.0 is interesting – can you say something about what conditions would cause the GWT to be inversely correlated with SATs?

5) Figure 3: It appears that the annual averaging is hiding important relations. That is, if surface air temperature (SAT) is the driver of groundwater temperature, it does not follow that the summary groundwater system temperatures would be warmer than the SATs at every location. Is it not likely winter periods skew the annual SAT, but the groundwater system is buffered from these colder temperatures? Therefore, might it be more insightful to look at SATs during non-winter conditions?

6) Lines 176-177: For this sentence: "This indicates that GWTs are often influenced by local causes and not necessarily solely by surface temperatures.", the correlation is between the weather station that is measuring surface temperatures correct? Then wouldn't it follow that the correlation is between groundwater temperatures and local SATs?

7) Lines 220-225: Can you provide reasons (and citations for the interested reader) for why there are different levels of change with land use?

8) Line 248: Did the hot spring suddenly appear or was it always there and something else changed? It was not apparent to me in Figure 5 what is the hot spring effect that I should be seeing in IIb and IIc in Figure 5. It does seem these outlier wells that have known atypical perturbations make the narrative hard to follow because they pop up every time a point is being made, and cause two sets of statistics to be reported – one with them and one without them (e.g., Villach wells, lines 265-359, wells near the Drau River). Because you know they are not representative of the larger scale climate driver would it not be clearer to just state this in the beginning and say you are not going to include them when reporting the subsequent statistics?

9) Please describe briefly the technique of Menberg et al. (2014) and define "regime shift index" used to save the reader from having to find it.

10) Lines 296-297 and 313-314: There are other statistical tests that beyond linear and regime shift methods (such as autoregressive integrated moving average techniques). Were any of these tried? The difference in RMSE is reported here is so small that it seems a stretch to say one performs superior than the other, and maybe other methods would perform better.

11) Is there something we can learn about the fact that nationwide correlation is higher than any of the individual weather station / well combinations? Would it be worth including a sentence in the manuscript pointing out that if researchers simply used the nationwide relation they could potentially hurt their ability to solve their more local problem?

Minor Comments / Technical Corrections

Line 19: It would be nice to relate the locations to features transferable to other parts of the world (e.g., high topographic relief/mountainous versus less topographic relief/less mountainous).

Lines 47-67: Bill Selbig used a regression of historical groundwater and air temperatures for the purpose of forecasting what future groundwater temperatures would be given expected changes calculated by GCMs. Not sure if your work would benefit from an application of how groundwater temperature trends influences societally relevant endpoints such as trout. There are others as well, but this work can be found in: Hunt, R.J., Walker, J.F., Selbig, W.R., Westenbroek, S.M, and Regan, R.S., 2013, Simulation of Climate-Change Effects on Streamflow, Lake Water Budgets, and Stream Temperature Using GSFLOW and SNTEMP, Trout Lake Watershed, Wisconsin: U.S. Geological Survey Scientific-Investigations Report 2013-5159, 118 p., http://pubs.usgs.gov/sir/2013/5159/.

Figure 1: the dashed line is not defined in the figure or in the caption.

Lines 158-159: It would be clearer to state exactly what is meant when stating

"...the distance in the north-south direction of two wells has more influence on the correlation...." As written the influence can be augmenting (more correlation) or degrading (less correlation).

Line 176: I don't think figure 3 shows "pairs of wells" but wells within 5 km of a weather station.

Lines 205-206: It seems Vienna may not be the best example to state as it only has one well included in its calculation of correlation.

Line 240-241: I am not sure I followed the sentence construction – what is meant by "...but in one sudden drop or rather rise in temperatures."?

Lines 222-224: In the beginning of this paragraph the topic is rate of change and then in these lines it is absolute change over a period, then the next paragraph goes back to rate of change. Perhaps better to start out with the differences in absolute temperatures then stay with changes in temperature. Also, the period 1990-2012 stated in these lines is not the same as reported in the caption of Figure S2 (01/1994 – 12/2013).

Line 239: Here is perhaps an opportunity to reinforce the importance of including groundwater flow when trying to interpret groundwater temperature (as opposed to dry borehole temperatures mentioned in the introduction). Same with line 304 in the Conclusions.

Line 240: Are there other cases of extreme changes not discussed in the text?

Line 243: The word "extend" should be "extent".

Line 236-249: The discussion starts with the <5% cases then includes the >95% then concludes again with <5%.

Line 261: My PDF had an odd "extend" tacked onto the end of the line.

Line 265-266: I think this sentence is less clear than it could be. I think the point is that if SATs are the driver of GWTs the former cannot lag behind the latter. The fact

that GWT changes precede the SAT driver suggests this method does not have the resolution to determine short lags between SATs and GWTs.

Line 303: "instalment" should be "installment", or even better, "installation"

Figure 6: I am not sure what to make of the checkerboard bar around 2006.

Figure S5: Perhaps add a vertical line to the figure to help the reader identify the exact date of the July 2007 flood.

---

## Author Comment (AC1) · 2 May 2018

**Response to Reviewer #1 of the manuscript**

**"Recent trends of groundwater temperatures in Austria"**

**by Benz et al. submitted to *Hydrology and Earth System Sciences*.**

Manuscript Number: hess-2017-663

Revision due before: 4 May 2018

**Reviewer comments:**

This paper addresses an important and interesting topic regarding the influence of atmospheric warming on groundwater temperature (GWT) in shallow systems. The authors used temperature records from 229 wells located in Austria and climatic data from weather stations installed nearby the wells. The positioning of the paper within the framework of studies devoted to the impact of climate change on hydrological system is well presented. The authors found that nationwide temperatures of groundwater increase and correlate statistically well with surface air temperature (SAT). Additionally, authors have used linear and step-wise models to describe the evolution of temperatures. Based on the step-wise approach (which seems to be more accurate than the linear model) the authors have identified that groundwater respond to climate regime shifts with sudden increase in temperature. This paper has been carefully prepared and is well written. The conclusions will definitely trigger the attention of the scientific community and the readers of HESS. Nevertheless I believe that some points need to be clarified before publication.

**Reply:** Thank you very much for your kind words and constructive comments. Your concerns are addressed in the following.

**Rev #1 General comments:**

**Rev #1, Comment # 1:** Some aspects of the methodology are not clear or absent. More details on how the 229 wells investigated in this study have been selected is required. More information regarding the type of sensors used to monitor GWT would be helpful to appreciate the quality of the data analyzed. More information regarding the regression approaches is also needed. How the shifts in regimes are determined in the step-wise model (mathematically speaking)? I also raise some additional points regarding the methodology in the specific comments.

**Reply:** We agree. While no specific information on the type of sensors used for monitoring is available to us, the information on the well selection strategy was extended: *"In Austria, GWTs up to Dec 2013 are provided by the Austrian Federal Ministry of Sustainability and Tourism Directorate-General IV. - Water Management (BMNT, former Federal Ministry of Agriculture, Forestry, Environment and Water Management (BMLUFUW)) in 1138 wells. Here, we focus on all wells with a measurement depth of less than 30 m, a record of at least 20 years and no major*

*breaks (> 3 month) in the last 20 years of the time series. Hence, all studied wells are monitored at least since Jan 1994, and some already since 1966 (see Fig. S1a for more information). Additionally wells impacted by geothermal hot springs were excluded. Overall in this study …"* (lines 109ff).

The paragraph discussing the linear analysis now reads (lines 152ff): "*Equivalent to the work by Lee et al. (2014), a linear temperature change was determined for all 227 wells. For this, a linear regression model of the annual mean temperature data was determined in Matlab 2016b. Because all wells in our dataset were continuously monitored between January 1994 and December 2013, only this timeframe was analysed.*"

Information on the step-wise model was also extended (lines 159ff). The mathematical basis of the method can be found in the given references: "*… in recent years the method by Rodionov (2004) became standard. It identifies the significance of each possible shift by calculating the so-called Regime Shift Index (RSI): the cumulative sum of the normalized differences between the observed values and the long term mean of the assumed regime. Only shifts with a positive RSI are considered significant, and a higher value of RSI denotes a more pronounced CRS. The entire algorithm is described in detail by Rodionov (2004). This sequential analysis is data driven and requires no prior knowledge of the timing of possible shifts. It was updated to further include prewithening in order to reduce background noise (Rodionov 2006) and is available online as a Microsoft Excel add-in (NOAA). In this study we applied the method to the complete timeseries of all 227 wells and 12 weather stations. Because the algorithm cannot handle gaps within the analysed series, gaps in our data were filled using a linear fit …*"

**Rev #1, Comment #2:** I believe that there is a discrepancy between the original objective of the paper, which aims at highlighting impact of climate change at regional (country) scale (Line 12), and the description of potential local effects for (some) specific wells and locations. Indeed, the authors describe potential factors which could explain uncorrelated data locally. Local information that are made available to the reader are to my opinion not sufficiently detailed to support the arguments. The conclusions are consequently difficult to trust. I would recommend to separate the description of local factors from the result of the regional statistical analysis (which to my opinion constitutes the novelty of this study). The local impacts could be introduced in a separate discussion section. In this specific section, the authors could provide an exhaustive list of potential factors that could explain uncorrelated data along with some examples from specific sites to illustrate the hypothesis.

**Reply:** We agree and separated local and countrywide results more clearly in the Results chapter. Now all subsections (correlation, linear fit, and climate regime shift) have a countrywide discussion first and a more local discussion second. Additional detail for each specific location such as the immediate surrounding, land use, or similar is now given where applicable.

**Rev #1, Comment #3:** I believe that the conclusions of this paper could be strengthened by performing a more robust multivariate statistical analysis (Principal Component Analysis for example) considering more factors which might have an influence on GWT, integrating not only SAT but also geology, land cover evolution, water level variation, precipitation, population dynamic, length of the temperature time series:

**Reply:** Thank you very much for this comment, this would be very interesting indeed and we hope to implement this in future studies. However, the suggested analysis is far beyond the scope of this manuscript and most of the mentioned parameters such as geology, land cover evolution, water level variation and population dynamic, are not available to us yet. If you are interested in this topic please feel free to contact us at susanne.benz@kit.edu.

**Rev #1 Specific comments**

**Rev #1, Comment # 4:** Line 68: "...over decades". Please be more precise here.

**Reply:** The text was changed to *"...GWTs of 227 wells in Austria, measured in part since 1966, are analysed ..."* (line 80).

**Rev #1, Comment # 5:** Line 73: "... step-wise increases between the regimes". This is not clear to me. What regimes? Please clarify.

**Reply:** We agree. Text was changed to: *"These control atmospheric temperatures as well and are often described as sudden, step-wise temperature changes separating stable periods, called climate regimes."* (lines 85ff).

**Rev #1, Comment # 6:** Figure 1 b. needs clarification. The presence of 3 curves is confusing. Could you, for example, make the inner percentile filled with transparent colors?

**Reply:** We agree. The dashed line was used to show the $95^{th}$ percentile, but transparent color is a better idea. We changed it accordingly:

[Figure]

**Figure 1. (a) Location of all analysed groundwater temperature (GWT - 227 wells) and surface air temperature (SAT - 12 weather stations) measurement points; (b) temporal evolution of the spatial median, annual mean temperatures for groundwater (blue) and air (red). The inner 90 percentiles are marked in lighter colours. All time series were monitored since at least 1994.**

**Rev #1, Comment # 7:** Line 98: How the wells have been selected? What proportion of wells has been excluded from the database? See general comment.

**Reply:** Information on how the wells where selected is now given in the manuscript. See Rev #1, Comment #1 for details.

**Rev #1, Comment # 8:** Line 128. Please clarify why you choose 1994 as initial time for fitting.

**Reply:** A clarifying sentence was included: "*Because all wells in our dataset were continuously monitored between 1994 and 2013, only this timeframe was analysed.*" (lines 152f).

**Rev #1, Comment # 9:** Line 129: Knowing which software you used is not informative here...

**Reply:** We agree. It is not necessary to know, but as it is common curtsey to give this information, we would like to keep it.

**Rev #1, Comment # 10:** Line 132: Please justify the choice of using the Spearman correlation coefficient and provide references.

**Reply:** Spearman correlation was chosen as it is more robust to outliers than other correlation measures. A sentence was added for clarification (lines 145f): *"Within this study, the Spearman correlation coefficient was determined, as it is especially robust to outliers caused for example by heat waves, which impact air temperatures but have only minor effect on groundwater temperatures."*

**Rev #1, Comment # 11:** Line 133: Taking annual mean values calculated with 8 months of data only may introduce some bias... Considering only years with full year of data would be more robust to my opinion. Otherwise, please discuss the limitations in the text. It is also not clear why yearly averages are used in the correlation analysis while the linear regressions are performed on monthly mean temperature (Line 129).

**Reply:** We agree, the linear analysis was changed and is now working with annual mean data as well. Interestingly, this decreased the determined temperature change for both GWT and SAT by about 0.1 K per 10 years compared to the analysis with monthly mean data. So far we are not certain what causes this discrepancy.
Additionally, the process of getting annual mean data was also revised, and gaps in the time series are now filled before determining the annual mean. The procedure is described in the chapter "Groundwater Temperatures" (lines 114ff): *"Overall, in this study annual mean data of 227 individual wells from all over the country (Fig. 1a) are analysed. Years with less than 9 months of data are excluded. For the timeframe 1994 and 2013, this amounts to 74 excluded data points in 60 wells. Additionally, only 9-11 months of data were available for 260 data points in 122 wells. To minimize the associated bias, these small gaps in the time series were filled using a linear fit. Hence small errors for years without a full set of monthly mean data have to be expected."*

**Rev #1, Comment # 12:** Line 132-136: It would be interesting to perform complementary correlation analysis accounting for other parameters such as depth of the wells, depth to the water table, geology, vegetation and land use. This could be assess with multivariate methods such as PCA. This could add valuable picture of the factors influencing the results.

**Reply:** We generally agree, however this analysis is beyond the scope of this manuscript. Please find our previous reply to Comment # 3 for more information.

**Rev #1, Comment # 13:** Line 145: "Breaks within the data were filled using linear fit". This is not clear... Please provide more information why you have to fill gaps for this analysis (and not for the other analysis?).

**Reply:** More Information on of the climate regime shift analysis is now given in the manuscript. See Rev #1, Comment #1 for details. This includes further information on why gaps have to be filled: *"In this study we applied the method to the complete time series of all 227 wells and 12 weather stations. Because the algorithm cannot handle*

*gaps within the analysed series, gaps in our data were filled using a linear fit.*" (Lines 166ff).

**Rev #1, Comment # 14:** Lines 160-164 and Figure 2a and b: This part require clarifications. As the authors stated, it seems that the shape of Austria (political boundary) might influence the results. Also the topography, with E-W strike orientation, might also have an influence. It is not so surprising that the correlation is better E-W that N-S (same latitude and orientation of topography). I am wondering if the figures are really informative...the decreasing correlation with distance in the figure a) is not obvious with the sharp increase at 550 km... Does this distance correspond to a decrease of the number of wells considered in the calculation?

**Reply:** We agree. Hence, we changed the entire paragraph to clarify this issue (lines 183ff): "*Additionally, the correlation between two wells seems to be anisotropic: correlation coefficients between two wells decrease faster with north-south distance than with west-east distance (Fig. 2b), which can be explained by the dominant striking direction of the geology and the resulting topography in Austria, where valleys generally run from west to east. Hence, larger rivers typically follow this direction and wells at the same latitude experience similar temperature signals.*"

Additionally the number of pairs of wells is now also given for each distance in Figure 2a):

[Figure]

**Figure 2. Influence of distance on the correlation between the annual means of two measurement points. a) Correlation between SAT time series is given in red, median correlation between GWT time series is given in blue. The inner 90 percentile are coloured in grey, the number of pairs of wells per distance is shown in dark blue below. b)**

**The colour gives the median correlation between GWTs of two wells in relation to their absolute distance to each other in east-west direction (x-axis) and in north-south direction (y-axis).**

**Rev #1, Comment # 15:** Figure 3 is interesting but difficult to read. Would it be clearer if you display the relative change in temperature for all the wells? What are the p values here (not introduced in the text)?

**Reply:** We agree, the figure gives now relative temperature change, and p-values are provided:

[Figure]

**Figure 3. Change from 1994 in surface air temperature (SAT) and groundwater temperatures (GWTs) of all wells within 5 km of the analysed weather station. See Fig. S3 for an overview of the locations. Minimum and maximum correlations and p-values between individual wells and weather stations are given.**

**Rev #1, Comment # 16:** Line 175 -176: To what coefficient are you referring to? The p values in the figure 3?

**Reply:** It is the Spearman correlation coefficient. For clarification, we changed the sentence as follows: *"... and Spearman correlation coefficients are < 0.5 ..."* (Lines 201f).

**Rev #1, Comment # 17:** Lines 187-190: Here it seems that the length of the time series is critical in the interpretation of the correlation analysis... Please discuss this point.

**Reply:** The discussion of the length of the time series was extended as follows: *"The well with the highest correlation of 0.80 to SAT is located less than 1 km from the weather station close to the airport parking lot next to suburban housing. It is continuously monitored since 1970 and the longest time series in the area. The well with the lowest correlation (0.45) to the weather station here is located slightly to the east near a dog-park and suburban housing. Here observations started in 1994, it is the shortest time series in this area. At all other wells, measurements began in 1986 and show correlations between 0.6 and 0.7 to SAT indicating that the duration of the measurements play a significant role for local comparisons. In contrast, duration of the time series appears to be of minor importance on a countrywide scale. For example, the long time series in Wiener Neustadt (Fig. 3), which started measurements in 1970 and is located near a mineral extraction site, has a correlation of 0.48 and is therefore comparable to the short time series in Graz, starting in 1994 located in a suburban area."* (Lines 212ff).

**Rev #1, Comment # 18:** Table 1: What does p-value mean here? Not introduced in the text or the caption....

**Reply:** Sorry for being unclear. Those are the p-values of the correlations. An explanation was added in the table caption: *"Correlation coefficient and corresponding p-value between spatial median SAT and spatial median GWT for all analysed SAT locations and additional information "*

**Rev #1, Comment # 19:** Lines 205-206: Reference to table is missing. It is actually not a big difference of correlation coefficient 0.36 vs 0.24... The comparison with population density is not obvious to me from these values. Please clarify. The influence of city center and development of urban area is actually critical. Could it be possible that the increases in temperatures are partly related to urban development? Identifying the correlation with such factors could be assessed with a multivariate correlation methods (PCA).

**Reply:** We agree. Hence, a reference to the table was added and the paragraph (lines 234ff) was changed following also the recommendation by Rev. #2:
*"In addition, the data indicates that city size or rather population of the city does not necessarily influence the correlation between GWT and SAT (Table 1). For example, both locations Graz (population of more than 250,000) and Eisenstadt (population of 13,000) have similar correlation coefficients despite their vastly different population. Meanwhile, Bregenz and Feldkirch have a similar population (~30,000) and number of wells (six), but different correlation coefficients (0.52 and 0.19). However, it is also important to note that not all wells analysed here are located in the city centre, still all of them are within close proximity (< 250 m) of build-up and urban areas (Fig. S3)"*

Regarding the second part of your comment: The comparison of temperature increase and land cover type in Figure S2b indicates that there is no link between land cover (such as artificial surface and forest) and temperature change on the scale analysed here.

**Rev #1, Comment # 20:** Line 214-215: This difference in average changes in temperature with higher values for GWT than SAT is surprising… Could it reflect the effect of urban development or other anthropogenic activities (pumping, injection, heating system...).

**Reply:** Yes, it is not as expected. However, it is not only observed in urban areas, but also in rural areas. Our current hypothesis for slightly higher GWT increase than SAT increase are due to the chosen timeframe 1994 to 2013. In 1994 there was a heat wave over Central Europe and annual SAT was considerable higher than the long-term average at that time. The text was extended to include this discussion (lines 248ff): *"During the time between 1994 and 2013, GWTs have changed on average by +0.36 ± 0.44 K per 10 years and SAT on average by +0.24 ± 0.13 K per 10 years. The lower changes in SAT are most likely due to the chosen timeframe: A heat wave in summer 1994 led to extraordinary high annual mean SAT in this year (Figure 1b) and thus impacts the determined linear temperature change."*

**Rev #1, Comment # 21:** Line 226: Please provide a reference to the figure supporting the statement that spatial pattern of temperature changes is visible...

**Reply:** Reference to Fig. 4 was added (line 266).

**Rev #1, Comment # 22:** Lines 226-235: Too few information are available on the effect of this flood event. What was the difference in temperature between the river and GW during the event? Did it cover the entire well area? Estimated volume? Please provide more information or I would recommend to remove this paragraph.

**Reply:** We agree and removed any discussion of the flooding as insufficient information is available to prove a link between both events.

**Rev #1, Comment # 23:** Line 236-249: It is somehow surprising and confusing how local effects are introduced again... I believe that it should be discussed in a dedicated section discussing potential hypothesis that may explain uncorrelated data with eventually some examples of local factors from specific sites as examples.

**Reply:** We agree, the Results chapter was restructured, and countrywide and local factors are now discussed separately. See our reply to your comment #2 for more information.

**Rev #1, Comment # 24:** Line 247-249: Do you mean that the hot springs appeared suddenly?... I imagine that they were active before and constitute a constant temperature boundary...

**Reply:** The hot springs are known since the roman ages and their touristic use goes back to much earlier times than the beginning of the herein used monitoring data. We assume

there was some construction work or something similar going on and hydrogeological conditions changed. However, we could not find concrete evidence of this. Following Rev #2 Comment #2, all wells dominated by this hot spring are now taken out of the analysis.

**Rev #1, Comment # 25:** Line 262: I do not understand what the authors mean by "spatial median annual mean"... please clarify.

**Reply:** We agree. The sentence was simplified as follows: *"All detected climate regime shifts (CRS) of the spatial median temperature time series are shown in Fig. 6a.."* (Line 293).

**Rev #1, Comment # 26:** Lines 263 - 266: I am confused here. How do you explain that the shift in GWT occurs earlier than for the SAT? If the "CRS method (do you mean step-wise method) cannot be used to determine the precise timelag between GWT and SAT" why do you use it?

**Reply:** Yes, the original sentence was unclear. The CRS method was previously used to determine the time lag between GWT and SAT, but our results indicate that it is actually not precise enough to do so. The sentence was therefore changed following a suggestion also by Rev #2 Comment #25: *" ... the shift in the late 90s appears earlier and is more significant in GWTs. However, because SATs are the drivers of GWTs and not vice versa, the fact that the GWT change precedes the SAT change suggests that this method does not have the necessary resolution to determine short time lags between SATs and GWTs. Accordingly ... "* (Lines 298ff).

**Rev #1, Technical corrections:**

**Rev #1, Comment # 27:** Line 29-31: Reference is missing.

**Reply:** Two references were added: *"While, already at depth of a few meters, the amplitudes of periodic diurnal and seasonal temperature trends are strongly attenuated (Taylor and Stefan, 2009), long term non-periodic changes of air temperature permanently influence the subsurface down to greater depths of several tens to hundreds of meters (Beltrami et al., 2005)."* (Lines 30ff).

Beltrami, H., Ferguson, G., and Harris, R. N.: Long-term tracking of climate change by underground temperatures, Geophysical Research Letters, 32, 1–4, 2005.
Taylor, C. A. and Stefan, H. G.: Shallow groundwater temperature response to climate change and urbanization, Journal of Hydrology, 375, 601–612, doi:10.1016/j.jhydrol.2009.07.009, 2009.

**Rev #1, Comment # 28:** Line 58: Reference style for Menberg et al. (2014).

**Reply:** Was updated from "*(Menberg et al., 2014)*" to "*Menberg et al. (2014)*" (line 69).

**Rev #1, Comment # 29:** Line 72: Check reference style.

**Reply:** Was changed to "*e.g. Minobe (1997) and Rodionov (2004)*" (line 84).

**Rev #1, Comment # 30:** Line 128: should be "Equivalent to the work by Lee et al. (2014)".

**Reply:** Sentence was changed from *"(Lee et al., 2014)"* to *"Lee et al. (2014)"* (line 151).

**Rev #1, Comment # 31:** Labels of figure 2b could be changed by Northing and Easting.

**Reply:** To avoid any confusion labels of both axis were changed to "*distance between two wells in east-west / north-south direction [km]*".

**Rev #1, Comment # 32:** Figure 6. Please add legends to your figures.

**Reply:** Legend was added to Figure 6. Additionally the bar plot was changed to transparent following Rev #2, Comment #27.

[Figure]

**Figure 6. (a) Median groundwater temperature (blue) and surface air temperature (red) of all wells or rather weather stations as well as the corresponding climate regime shifts (CRS) in form of the regime shift index (RSI). (b) Percentage of measurement points in GWT (blue) and SAT (red) that show a CRS in each year. The analysis of global temperatures data indicates a regime shift at the end of the 70s, the 80s and the 90s which are shown here in as grey bars.**

---

## Author Comment (AC2) · 2 May 2018

**Response to Reviewer #2 of the manuscript**

**"Recent trends of groundwater temperatures in Austria"**

**by Benz et al. submitted to *Hydrology and Earth System Sciences*.**

Manuscript Number: hess-2017-663

Revision due before: 4 May 2018

**Reviewer comments:**

Thank you for the opportunity to review "Recent trends of groundwater temperatures in Austria" by Benz, Bayer, Winkler, and Blum. I enjoyed reading the manuscript and appreciate the work it represents. I have outlined my specific primary suggestions for improvement below. I've also included minor comments, along with typographical suggestions as requested by the Journal. Only the primary comments rise to the level of serious consideration and response. The authors should feel free to contact me if anything is unclear at rjhunt@usgs.gov.

**Reply:** Thank you very much for your kind words.

**Rev #2 Specific/Primary Comments:**

**Rev #2, Comment # 1:** Overall manuscript: It strikes me that a focus on annual air temperature misses a fundamental process important for this discussion. The temperature of the groundwater system reflects the temperature of groundwater recharge. Groundwater recharge, however, is variable over time, thus annual temperature changes are likely too coarse to capture the temperature effects of inter-annual recharge process. That is: snowmelt recharge will be near 0 degrees C; rain-derived recharge will be warmer. Perhaps there is a shift in recharge from less snowmelt to warmer rain sources that is driven by air temperatures. A groundwater recharge approach means that the simple relation of air temperature to groundwater temperature is more indirect, and this additional "noise" to the signal is perhaps why the correlations are not higher.

**Reply:** We agree that a more detailed consideration of the annually changing recharge temperature would improve our general understanding of the vertical heat transport process in the unsaturated zone. However, such data is not available to us and previous studies have indicated this to be of minor importance, and thus it is does not included in our analysis. Further discussion of recharge processes, citing relevant sources, was added to the introduction in lines 44ff:

*"When dynamic groundwater flow conditions exist, then advective heat transport can substantially affect the thermal regime in the subsurface [...]. Additionally, recharge processes, including snowmelt and rain-derived recharge, might impact the thermal regime of the shallow subsurface. Previous studies, however, indicate that in many cases their influence can be neglected. Ferguson and Woodbury (2005) and Bense and*

*Kurylyk (2017) demonstrated that it is possible to estimate groundwater recharge by using temperature-depth profiles based on the common assumption that the mean annual groundwater recharge temperature is equal to the mean annual surface air temperature. Menberg et al. (2014) showed in their study that the contribution of snowmelt-induced recharge with low temperature is minor in comparison to the overall recharge. Finally, Molina-Giraldo et al. (2011) investigated the impact of seasonal temperature signals into an aquifer upon bank infiltration including also varying groundwater recharge temperatures. They showed that the convective heat transfer by groundwater recharge compared to conduction through the unsaturated zone and convection within the aquifer is of minor impact. Still, the interplay of long-term climate variations, land use change and groundwater produces a complex transient system, which is difficult if not impossible to accurately understand based on a few borehole measurements."*

**Rev #2, Comment # 2:** Section Groundwater Temperature/Figure S1: Similar to comment #1, groundwater basins have a residence time, with multiple ages and potential lags. There is an assumption that all groundwater reflects current air temperatures (e.g., line 221) but this may not be the case. Given the importance of other factors such as residence time, and the unsaturated zone buffering that dampens the climatic drivers, it seems worthwhile to include well statistics relating to:

- Depth to water table
- Well open interval
- Distance the well's open interval is below land surface
- Distance the well's open interval is below the water table
- Estimated position in the groundwater flow system (e.g., uppermost, middle, discharge; near groundwater divide versus near flow system end; urban versus rural agriculture versus forest; high elevation versus low elevation)

**Reply:** We agree and have now included a discussion of temperature measurement depth in the chapters discussing correlation and linear temperature change. All wells are observation wells and are open all the way through. Unfortunately there is no information on the position in the groundwater flow system. Depth to the water table is also monitored in all wells, but currently not available to us.

The chapter discussing correlation between SAT and GWT now includes a discussion of measurement depth (lines 219ff): *"Additionally, measurement depths of GWT can have an impact on the correlation between SAT and GWT. While it is generally assumed that a measurement depth closer to the surface results in a better correlation with SAT as there is less of a shift between both datasets, this is only the case for some of the here analysed locations such as Villach (Figure S4a). In contrast, correlation increases with GWT measurement depth for other locations such as the one in Graz. This might be related to local underground heat sources such as sewage systems impacting GWT near the surface more than temperatures at greater depth. However, as the depth of the wells analysed here varies only slightly, no definite conclusions can*

*be drawn without further inspection of specific cases."*

Additionally, information on the GWT measurement depth for all wells next to weather stations is now included in Table 1.

**Table 1. Correlation coefficient and corresponding p-value between spatial median SAT and spatial median GWT for all analysed SAT locations, and additional information.**

| Location | Number of wells | Measurement depth GWT [m below surface] | Number of weather stations | Spearman correlation | p-value | Population[1] |
|---|---|---|---|---|---|---|
| Linz | 1 | 10 | 1 | -0.31 | $10^{-1}$ | 192,000 |
| Feldkirch | 6 | 4 to 17 | 1 | 0.19 | $10^{-1}$ | 31,000 |
| Innsbruck | 2 | 10 | 2 | 0.37 | $10^{-1}$ | 123,000 |
| Vienna | 1 | 12 | 1 | 0.41 | $10^{-2}$ | 1,740,000 |
| Zeltweg | 2 | 6 to 7 | 1 | 0.48 | $10^{-3}$ | 7,000 |
| Wiener Neustadt | 2 | 9 to 20 | 1 | 0.51 | $10^{-4}$ | 42,000 |
| Bregenz | 6 | 4 to 10 | 1 | 0.52 | $10^{-3}$ | 28,000 |
| Tulln an der Donau | 1 | 7 | 1 | 0.54 | $10^{-2}$ | 15,000 |
| Eisenstadt | 2 | 4 to 5 | 1 | 0.67 | $10^{-4}$ | 13,000 |
| Graz | 9 | 4 to 12 | 1 | 0.73 | $10^{-8}$ | 266,000 |
| Villach | 17 | 3 to 11 | 1 | 0.80 | $10^{-11}$ | 60,000 |

[1] Register-based Labour Market Statistics 2014, municipality level (Statistik Austria).

The following paragraph was added to the chapter on linear temperature change (lines 259ff): *"Temperature change decreases slightly with GWT measurement depth by approximately 0.015 K per 10 years per meter (Fig. S4b). This relationship can be related to deeper temperatures corresponding to earlier temperatures, when temperature increase was less severe. However, because the vast majority of temperatures are monitored at a depth of less than 15 m and show a high variability in linear temperature change, this number must be taken with caution. R² of the fit is only 0.02 and RMSE is 0.4 K."*

[Figure]

**Figure S4: a) Influence of measurement depth on Spearman correlation between GWT and nearby SAT measurements. Shown are results for all wells depicted in Fig. S3. b) Influence of measurement depth on observed change in temperature. The best fit implies a linear temperature change of 0.48 K per 10 years for a depth of 0 m and a decrease in temperature change by 0.015 K per 10 years for each additional meter between measurement point and surface.**

**Rev #2, Comment # 3:** Lines 104-106: It seems that only focusing on annual averages may limit the applicability of the insights. For example, for cold water fisheries it is usually the temperatures in the late summer – late fall that are important.

**Reply**: We agree. However, in order to keep the data analyzed in this study consistent, we would like follow advice from Rev. #1 and focus solely on one set of temperature data (annual or monthly means). Because the climate regime shift analysis works best with annual mean data, we prefer working with this. A more detailed analysis of seasonal variation would require an extensive investigation of the data and this is beyond the scope of this study.

**Rev #2, Comment # 4:** Figure 2: The shaded area and short-duration blue line dipping below y=0.0 is interesting – can you say something about what conditions would cause the GWT to be inversely correlated with SATs?

**Reply:** Fig 2a) shows correlation between different wells on the y-axis and the distance between those wells on the x-axis. All correlation coefficients close to or below zero all have a p-value of close to one, meaning these wells do not correlate. It is likely that at least one well in each of these pairs is influenced by other, local heat sources and not by surface temperatures. A sentence was added to the manuscript to clarify this (lines 183ff): "*For the weather station, each individual pair is shown by a red point, for GWTs, as there are many possible pairs of wells, the line gives the moving median (± 25 km) correlation of all pairs at the corresponding distances. The inner 90 percentiles are shown in grey, and correlation coefficients close to or below zero are*

*determined for several pairs of wells. However, here p-values are generally also close to one and GWTs do not correlate. This is most likely due to local heat sources impacting at least on well in these pairs."*

**Rev #2, Comment # 5:** Figure 3: It appears that the annual averaging is hiding important relations. That is, if surface air temperature (SAT) is the driver of groundwater temperature, it does not follow that the summary groundwater system temperatures would be warmer than the SATs at every location. Is it not likely winter periods skew the annual SAT, but the groundwater system is buffered from these colder temperatures? Therefore, might it be more insightful to look at SATs during non-winter conditions?

**Reply:** Yes, you are right; the annual average is simplifying the complicated short-term relationship between above ground and subsurface temperatures. One example, as you mention, are cold air temperatures in winter: several studies have shown that snow cover insulates groundwater temperatures during that time and annual mean GWT are therefore warmer than annual mean above ground temperatures. Still, as discussed in response to your comment #3 we would like work with annual mean data only for this study.

**Rev #2, Comment # 6:** Lines 176-177: For this sentence: "This indicates that GWTs are often influenced by local causes and not necessarily solely by surface temperatures.", the correlation is between the weather station that is measuring surface temperatures correct? Then wouldn't it follow that the correlation is between groundwater temperatures and local SATs?

**Reply:** Correct. Surface temperature was the wrong term and therefore this was changed to "*local SATs*" (line 202).

**Rev #2, Comment # 7:** Lines 220-225: Can you provide reasons (and citations for the interested reader) for why there are different levels of change with land use?

**Reply:** We meant to say that groundwater temperatures do not change significantly differently for the different land cover classes. We changed the paragraph to make this message clearer: *"There appears to be no significant influence of land cover on the observed temperature change (Fig. S2c). Median temperature change is approximately 0.4 ± 0.4 K per 10 years for groundwater under artificial surfaces and forest areas, and 0.3 ± 0.5 K per 10 years under cultivated areas."* (Lines 256ff).

**Rev #2, Comment # 8:** Line 248: Did the hot spring suddenly appear or was it always there and something else changed? It was not apparent to me in Figure 5 what is the hot spring effect that I should be seeing in IIb and IIc in Figure 5. It does seem these outlier wells that have known atypical perturbations make the narrative hard to follow because they pop up every time a point is being made, and cause two sets of statistics to be reported – one with them and one without them (e.g., Villach wells, lines 265-359, wells near the Drau River). Because you know they are not representative of the larger scale climate driver would it not

be clearer to just state this in the beginning and say you are not going to include them when reporting the subsequent statistics?

**Reply:** Yes, you are right, we have decided to exclude the two wells that are dominated by these hot springs and now only 227 wells are analyzed in total. The wells nearby and not impacted by the hot springs are still included.

The hot springs are known since the roman ages and their touristic use goes back to much earlier times than the beginning of the herein used monitoring data. But some changes in the hydrogeological conditions must have happened when additional wells started to pick up the signal. However, we could not identify any concrete evidence of these changes.

**Rev #2, Comment # 9:** Please describe briefly the technique of Menberg et al. (2014) and define "regime shift index" used to save the reader from having to find it.

**Reply:** The paragraph describing the method was extended accordingly to include a brief description (lines 158ff): *"… in recent years the method by Rodionov (2004) became standard. It identifies the significance of each possible shift by calculating the so-called Regime Shift Index (RSI): the cumulative sum of the normalized differences between the observed values and the long term mean of the assumed regime. Only shifts with a positive RSI are considered significant, and a higher value of RSI denotes a more pronounced CRS [Climate Regime Shift]."*

**Rev #2, Comment # 10:** Lines 296-297 and 313-314: There are other statistical tests that beyond linear and regime shift methods (such as autoregressive integrated moving average techniques). Were any of these tried? The difference in RMSE is reported here is so small that it seems a stretch to say one performs superior than the other, and maybe other methods would perform better.

**Reply:** No, we did not apply any ARIMA models yet. However, this is a very interesting thought that we will consider it in the future.

**Rev #2, Comment # 11:** Is there something we can learn about the fact that nationwide correlation is higher than any of the individual weather station / well combinations? Would it be worth including a sentence in the manuscript pointing out that if researchers simply used the nationwide relation they could potentially hurt their ability to solve their more local problem?

**Reply:** Yes. This is definitely the main message here that groundwater temperatures are dominated by local features. While a national average can give us important information on large scale trends and problems, local temperatures might behave differently. A sentence was therefore added to the conclusion: *"This reveals the extent in which groundwater temperatures are dominated by local events, groundwater flow, and the thermal properties of the surrounding. When solving local problems we can*

*therefore not recommend relying on average relationships valid on a nation scale.*"
(Lines 334ff).

**Rev #2, Minor Comments / Technical Corrections:**

**Rev #2, Comment # 12:** Line 19: It would be nice to relate the locations to features transferable to other parts of the world (e.g., high topographic relief/mountainous versus less topographic relief/less mountainous).

Reply: We agree, a sentence discussing the need for future work transferring these results to other regions has been added to the conclusion (lines 343ff): *"However, further research dedicated to other climate parameters such as permafrost and snowfall is necessary to validate these findings. Additionally, our observations made in Austria should be transferred to similar regions in the world testing the transferability of the presented results."*

**Rev #2, Comment # 13:** Lines 47-67: Bill Selbig used a regression of historical groundwater and air temperatures for the purpose of forecasting what future groundwater temperatures would be given expected changes calculated by GCMs. Not sure if your work would benefit from an application of how groundwater temperature trends influences societally relevant endpoints such as trout. There are others as well, but this work can be found in: Hunt, R.J., Walker, J.F., Selbig, W.R., Westenbroek, S.M, and Regan, R.S., 2013, Simulation of Climate-Change Effects on Streamflow, Lake Water Budgets, and Stream Temperature Using GSFLOW and SNTEMP, Trout Lake Watershed, Wisconsin: U.S. Geological Survey Scientific-Investigations Report 2013-5159, 118 p., http://pubs.usgs.gov/sir/2013/5159/.

**Reply:** Thank you for your suggestion. This study is very interesting. Thus, we added this reference (line 60).

**Rev #2, Comment # 14:** Figure 1: the dashed line is not defined in the figure or in the caption.

**Reply:** The dashed line represented the 95$^{th}$ percentile of SAT. However, following the suggestion by Rev #1, Comment # 6, a transparent color is now used to show the inner percentiles.

[Figure]

**Figure 1. (a)** Location of all analysed groundwater temperature (GWT - 227 wells) and surface air temperature (SAT - 12 weather stations) measurement points; **(b)** temporal evolution of the spatial median, annual mean temperatures for groundwater (blue) and air (red). The inner 90 percentiles are marked in lighter colours. All time series were monitored since at least 1994 (marked in grey).

**Rev #2, Comment # 15:** Lines 158-159: It would be clearer to state exactly what is meant when stating "...the distance in the north-south direction of two wells has more influence on the correlation...." As written the influence can be augmenting (more correlation) or degrading (less correlation).

**Reply:** We agree. The sentence was changed to: *"Additionally, the correlation between two wells seems to be anisotropic: correlation coefficients between two wells decrease faster with north-south distance than with west-east distance (Fig. 2b), which can be explained by the dominant striking direction of the geology and the resulting topography in Austria, where valleys generally run from west to east."* (Lines 182ff).

**Rev #2, Comment # 16:** Line 176: I don't think figure 3 shows "pairs of wells" but wells within 5 km of a weather station.

**Reply:** Correct. The sentence was therefore changed to "*Here correlations vary greatly and Spearman correlation coefficients are < 0.5 for about half of all wells within 5 km of a weather station.*" (Lines 199f).

**Rev #2, Comment # 17:** Lines 205-206: It seems Vienna may not be the best example to state as it only has one well included in its calculation of correlation.

**Reply:** We agree, another example was added: *"For example, both locations Graz (population of more than 250,000) and Eisenstadt (population of 13,000) have similar correlation coefficients despite their different population. Meanwhile, Bregenz and Feldkirch have a similar population (~30,000) and number of wells (six), but different correlation coefficients (0.52 and 0.19)."* (Lines 238ff).

**Rev #2, Comment # 18:** Line 240-241: I am not sure I followed the sentence construction – what is meant by "...but in one sudden drop or rather rise in temperatures."?

**Reply:** Sentence was change for clarification: *"In general, most of the extreme changes in temperature appear to be linked to local causes and do not happen gradually, but rather rapidly over the short time span of one or two years."* (Lines 280f).

**Rev #2, Comment # 19:** Lines 222-224: In the beginning of this paragraph the topic is rate of change and then in these lines it is absolute change over a period, then the next paragraph goes back to rate of change. Perhaps better to start out with the differences in absolute temperatures then stay with changes in temperature. Also, the period 1990-2012 stated in these lines is not the same as reported in the caption of Figure S2 (01/1994 – 12/2013).

**Reply:** It appears that the wording of the paragraph was misleading. The last part discussed not temperature change over time, but the absolute difference between different time series. This discussion is completely independent of our main results and was now moved to the Materials and Methods section when introducing the groundwater dataset (lines 124ff):

*"Following the CORINE Land Cover (CLC) data from 2012 (Fig. S2a), 45 % of all wells are under artificial surfaces, 46 % under agricultural areas, and 9 % under forest following the 100 m × 100 m classification. In addition, CLC from 1990 was consulted, however, no land cover changes near any of the analysed wells are observed. Overall, for the time period 1994 – 2013 when all wells were monitored, absolute GWTs under artificial surfaces are on average 1.5 ± 0.3 K warmer than GWTs under forest; GWTs under agricultural areas are on average 0.6 ± 0.2 K warmer than GWTs under forest (Fig. S2b). This validates previous findings by Benz et al. (2017b) for GWTs in Germany, who identified even larger differences of up to 3 K between the individual land cover classes."*

[Figure]

**Figure S2. a) Corine Land Cover 2012 of Austria. None of the analyzed wells and weather stations experienced a land cover change since 1990. b) Spatial median GWTs for each of the individual land cover classes. All wells are monitored since at least 1994. c) Relationship between land cover and groundwater temperature (GWT) change between 1994 and 2013. There appears to be no significant influence.**

**Rev #2, Comment # 20:** Line 239: Here is perhaps an opportunity to reinforce the importance of including groundwater flow when trying to interpret groundwater temperature (as opposed to dry borehole temperatures mentioned in the introduction). Same with line 304 in the Conclusions.

**Reply:** We agree. The following sentence was added (lines 278ff): *"These wells seem to be affected by the new drinking water supply (four wells with a total pumping rate of about 100 l/s) located about 1 km in the south. This demonstrates the importance of including groundwater flow when trying to interpret groundwater temperature."* and in lines 331f:" *This reveals the extent in which groundwater temperatures are dominated by local events, groundwater flow, and the thermal properties of the surrounding.*"

**Rev #2, Comment # 21:** Line 240: Are there other cases of extreme changes not discussed in the text?

**Reply:** Interesting point. We checked this issue and found 57 wells with extreme changes (> 400% of average change per year). However, a more detailed analysis is beyond the scope of this study.

Rev #2, Comment # 22: Line 243: The word "extend" should be "extent".

**Reply:** Word was changed (line 283)

**Rev #2, Comment # 23:** Line 236-249: The discussion starts with the <5% cases then includes the >95% then concludes again with <5%.

**Reply:** The chapter was slightly restructured, it now starts with the <5% and ends with the >95% cases (lines 266ff).

**Rev #2, Comment # 24:** Line 261: My PDF had an odd "extend" tacked onto the end of the line.

**Reply:** We checked this issue, however cannot find any anomalies in our PDF version.

**Rev #2, Comment # 25:** Line 265-266: I think this sentence is less clear than it could be. I think the point is that if SATs are the driver of GWTs the former cannot lag behind the latter. The fact that GWT changes precede the SAT driver suggests this method does not have the resolution to determine short lags between SATs and GWTs.

**Reply:** Thanks for your suggestion. The sentence was changed: *"… , the shift in the late 90s appears earlier and is more significant in GWTs. However, because SATs are the drivers of GWTs and not vice versa, the fact that the GWT change precedes the SAT change suggests that this method does not have the necessary resolution to determine short time lags between SATs and GWTs. Accordingly …"* (Lines 298ff).

**Rev #2, Comment # 26:** Line 303: "instalment" should be "installment", or even better, "installation"

**Reply:** Changed (line 333).

**Rev #2, Comment # 27:** Figure 6: I am not sure what to make of the checkerboard bar around 2006.

**Reply:** In 2007 bars for SAT and GWT were at the exact same height. However, this changed when two wells in Villach were excluded (your comment #8). Still, we changed the bars to be transparent in order to clarify this issue. In addition, a legend was added to the figure following Rev #1, Comment #32.

[Figure]

**Figure 6. (a) Median groundwater temperature (blue) and surface air temperature (red) of all wells or rather weather stations as well as the corresponding climate regime shifts (CRS) in form of the regime shift index (RSI). (b) Percentage of measurement points in GWT (blue) and SAT (red) that show a CRS in each year. The analysis of global temperatures data indicates a regime shift at the end of the 70s, the 80s and the 90s which are shown here in as grey bars.**

**Rev #2, Comment # 28:** Figure S5: Perhaps add a vertical line to the figure to help the reader identify the exact date of the July 2007 flood.

**Reply:** Following a suggestion by Reviewer #1 all references to the flood were deleted. There is currently not enough information available (e.g. extend of the flooding) to proof our hypotheses that flood and drop in temperatures are related. Still, the year in question was marked in Figure S5 to make the figure clearer.

[Figure]

**Figure S5. Location of the Drava river, the groundwater monitoring wells around it and measurement stations within the river (EHYD, 2017). Also shown is the groundwater time series of all wells within 1 km of the river and all measured river parameters. While GWTs show a sudden drop in 2007 (marked in red), observed river parameters give no indication of an abnormal event around that time.**

---

## Author Response (AR1)

**Response to the Editor of the manuscript**

**"Recent trends of groundwater temperatures in Austria"**

**by Benz et al. submitted to** *Hydrology and Earth System Sciences*.

Manuscript Number: hess-2017-663

Revision due before: 15 May 2018

**Editor comment:**

The two reviews offer detailed points that require careful consideration and appropriate clarifications in a revised manuscript.
The authors' responses appear well thought out and convincing, and a revised manuscript reflecting the additional information and clarifications can fully address the reviews. Please proceed with a revision.

**Reply:** Thank you very much for editing our manuscript! In the following, please find the revised manuscript with marked changes.

[revised manuscript text omitted]